# Deep Sets

**Manzil Zaheer**[1,2]**, Satwik Kottur**[1]**, Siamak Ravanbhakhsh**[1]**,**
**Barnabás Póczos**[1]**, Ruslan Salakhutdinov**[1]**, Alexander J Smola**[1,2]
[1] Carnegie Mellon University      [2] Amazon Web Services
{manzilz,skottur,mravanba,bapoczos,rsalakhu,smola}@cs.cmu.edu

## Abstract

We study the problem of designing models for machine learning tasks defined on *sets*. In contrast to traditional approach of operating on fixed dimensional vectors, we consider objective functions defined on sets that are invariant to permutations. Such problems are widespread, ranging from estimation of population statistics [1], to anomaly detection in piezometer data of embankment dams [2], to cosmology [3, 4]. Our main theorem characterizes the permutation invariant functions and provides a family of functions to which any permutation invariant objective function must belong. This family of functions has a special structure which enables us to design a deep network architecture that can operate on sets and which can be deployed on a variety of scenarios including both unsupervised and supervised learning tasks. We also derive the necessary and sufficient conditions for permutation equivariance in deep models. We demonstrate the applicability of our method on population statistic estimation, point cloud classification, set expansion, and outlier detection.

## 1  Introduction

A typical machine learning algorithm, like regression or classification, is designed for fixed dimensional data instances. Their extensions to handle the case when the inputs or outputs are permutation invariant sets rather than fixed dimensional vectors is not trivial and researchers have only recently started to investigate them [5–8]. In this paper, we present a generic framework to deal with the setting where input and possibly output instances in a machine learning task are sets.

Similar to fixed dimensional data instances, we can characterize two learning paradigms in case of sets. In **supervised learning**, we have an output label for a set that is invariant or equivariant to the permutation of set elements. Examples include tasks like estimation of population statistics [1], where applications range from giga-scale cosmology [3, 4] to nano-scale quantum chemistry [9].

Next, there can be the **unsupervised setting**, where the "set" structure needs to be learned, *e.g.* by leveraging the homophily/heterophily tendencies within sets. An example is the task of set expansion (a.k.a. audience expansion), where given a set of objects that are similar to each other (*e.g.* set of words {*lion, tiger, leopard*}), our goal is to find new objects from a large pool of candidates such that the selected new objects are similar to the query set (*e.g.* find words like *jaguar* or *cheetah* among all English words). This is a standard problem in similarity search and metric learning, and a typical application is to find new image tags given a small set of possible tags. Likewise, in the field of computational advertisement, given a set of high-value customers, the goal would be to find similar people. This is an important problem in many scientific applications, *e.g.* given a small set of interesting celestial objects, astrophysicists might want to find similar ones in large sky surveys.

**Main contributions.**   In this paper, (i) we propose a fundamental architecture, *DeepSets*, to deal with sets as inputs and show that the properties of this architecture are both necessary and sufficient (Sec. 2). (ii) We extend this architecture to allow for conditioning on arbitrary objects, and (iii) based on this architecture we develop a *deep network* that can operate on sets with possibly different sizes (Sec. 3). We show that a simple parameter-sharing scheme enables a general treatment of sets within supervised and semi-supervised settings. (iv) Finally, we demonstrate the wide applicability of our framework through experiments on diverse problems (Sec. 4).

## 2 Permutation Invariance and Equivariance

### 2.1 Problem Definition

A function $f$ transforms its domain $\mathcal{X}$ into its range $\mathcal{Y}$. Usually, the input domain is a vector space $\mathbb{R}^d$ and the output response range is either a discrete space, e.g. $\{0, 1\}$ in case of classification, or a continuous space $\mathbb{R}$ in case of regression. Now, if the input is a set $X = \{x_1, \ldots, x_M\}, x_m \in \mathfrak{X}$, *i.e.*, the input domain is the power set $\mathcal{X} = 2^{\mathfrak{X}}$, then we would like the response of the function to be "indifferent" to the ordering of the elements. In other words,

**Property 1** *A function $f : 2^{\mathfrak{X}} \to \mathcal{Y}$ acting on sets must be permutation **invariant** to the order of objects in the set,* i.e. *for any permutation $\pi$ : $f(\{x_1, \ldots, x_M\}) = f(\{x_{\pi(1)}, \ldots, x_{\pi(M)}\})$.*

In the supervised setting, given $N$ examples of of $X^{(1)}, \ldots, X^{(N)}$ as well as their labels $y^{(1)}, \ldots, y^{(N)}$, the task would be to classify/regress (with variable number of predictors) while being permutation invariant w.r.t. predictors. Under unsupervised setting, the task would be to assign high scores to valid sets and low scores to improbable sets. These scores can then be used for set expansion tasks, such as image tagging or audience expansion in field of computational advertisement. In *transductive* setting, each instance $x_m^{(n)}$ has an associated labeled $y_m^{(n)}$. Then, the objective would be instead to learn a permutation **equivariant** function $\mathbf{f} : \mathfrak{X}^M \to \mathcal{Y}^M$ that upon permutation of the input instances permutes the output labels, *i.e.* for any permutation $\pi$:

$$\mathbf{f}([x_{\pi(1)}, \ldots, x_{\pi(M)}]) = [f_{\pi(1)}(\mathbf{x}), \ldots, f_{\pi(M)}(\mathbf{x})] \tag{1}$$

### 2.2 Structure

We want to study the structure of functions on sets. Their study in total generality is extremely difficult, so we analyze case-by-case. We begin by analyzing the **invariant** case when $\mathfrak{X}$ is a countable set and $\mathcal{Y} = \mathbb{R}$, where the next theorem characterizes its structure.

**Theorem 2** *A function $f(X)$ operating on a set $X$ having elements from a countable universe, is a valid set function,* i.e., ***invariant** to the permutation of instances in $X$, iff it can be decomposed in the form $\rho\left(\sum_{x \in X} \phi(x)\right)$, for suitable transformations $\phi$ and $\rho$.*

The extension to case when $\mathfrak{X}$ is uncountable, like $\mathfrak{X} = \mathbb{R}$, we could only prove that $f(X) = \rho\left(\sum_{x \in X} \phi(x)\right)$ holds for sets of fixed size. The proofs and difficulties in handling the uncountable case, are discussed in Appendix A. However, we still conjecture that exact equality holds in general.

Next, we analyze the **equivariant** case when $\mathfrak{X} = \mathcal{Y} = \mathbb{R}$ and $\mathbf{f}$ is restricted to be a neural network layer. The standard neural network layer is represented as $\mathbf{f}_{\Theta}(\mathbf{x}) = \boldsymbol{\sigma}(\Theta\mathbf{x})$ where $\Theta \in \mathbb{R}^{M \times M}$ is the weight vector and $\sigma : \mathbb{R} \to \mathbb{R}$ is a nonlinearity such as sigmoid function. The following lemma states the necessary and sufficient conditions for permutation-equivariance in this type of function.

**Lemma 3** *The function $\mathbf{f}_{\Theta} : \mathbb{R}^M \to \mathbb{R}^M$ defined above is permutation **equivariant** iff all the off-diagonal elements of $\Theta$ are tied together and all the diagonal elements are equal as well. That is,*

$$\Theta = \lambda\mathbf{I} + \gamma\left(\mathbf{1}\mathbf{1}^{\mathsf{T}}\right) \qquad \lambda, \gamma \in \mathbb{R} \quad \mathbf{1} = [1, \ldots, 1]^{\mathsf{T}} \in \mathbb{R}^M \qquad \mathbf{I} \in \mathbb{R}^{M \times M} \text{ is the identity matrix}$$

This result can be easily extended to higher dimensions, *i.e.*, $\mathfrak{X} = \mathbb{R}^d$ when $\lambda, \gamma$ can be matrices.

### 2.3 Related Results

The general form of Theorem 2 is closely related with important results in different domains. Here, we quickly review some of these connections.

**de Finetti theorem.** A related concept is that of an exchangeable model in Bayesian statistics, It is backed by deFinetti's theorem which states that any exchangeable model can be factored as

$$p(X|\alpha, M_0) = \int \mathrm{d}\theta \left[\prod_{m=1}^{M} p(x_m|\theta)\right] p(\theta|\alpha, M_0), \tag{2}$$

where $\theta$ is some latent feature and $\alpha, M_0$ are the hyper-parameters of the prior. To see that this fits into our result, let us consider exponential families with conjugate priors, where we can analytically calculate the integral of (2). In this special case $p(x|\theta) = \exp\left(\langle\phi(x), \theta\rangle - g(\theta)\right)$ and $p(\theta|\alpha, M_0) = \exp\left(\langle\theta, \alpha\rangle - M_0 g(\theta) - h(\alpha, M_0)\right)$. Now if we marginalize out $\theta$, we get a form which looks exactly like the one in Theorem 2

$$p(X|\alpha, M_0) = \exp\left(h\left(\alpha + \sum_m \phi(x_m), M_0 + M\right) - h(\alpha, M_0)\right). \tag{3}$$

**Representer theorem and kernel machines.** Support distribution machines use $f(p) = \sum_i \alpha_i y_i K(p_i, p) + b$ as the prediction function [8, 10], where $p_i, p$ are distributions and $\alpha_i, b \in \mathbb{R}$. In practice, the $p_i, p$ distributions are never given to us explicitly, usually only i.i.d. sample sets are available from these distributions, and therefore we need to estimate kernel $K(p, q)$ using these samples. A popular approach is to use $\hat{K}(p, q) = \frac{1}{MM'} \sum_{i,j} k(x_i, y_j)$, where $k$ is another kernel operating on the samples $\{x_i\}_{i=1}^M \sim p$ and $\{y_j\}_{j=1}^{M'} \sim q$. Now, these prediction functions can be seen fitting into the structure of our Theorem.

**Spectral methods.** A consequence of the polynomial decomposition is that spectral methods [11] can be viewed as a special case of the mapping $\rho \circ \phi(X)$: in that case one can compute polynomials, usually only up to a relatively low degree (such as $k = 3$), to perform inference about statistical properties of the distribution. The statistics are exchangeable in the data, hence they could be represented by the above map.

# 3 Deep Sets

## 3.1 Architecture

**Invariant model.** The structure of permutation invariant functions in Theorem 2 hints at a general strategy for inference over sets of objects, which we call DeepSets. Replacing $\phi$ and $\rho$ by universal approximators leaves matters unchanged, since, in particular, $\phi$ and $\rho$ can be used to approximate arbitrary polynomials. Then, it remains to learn these approximators, yielding in the following model:
- Each instance $x_m$ is transformed (possibly by several layers) into some representation $\phi(x_m)$.
- The representations $\phi(x_m)$ are added up and the output is processed using the $\rho$ network in the same manner as in any deep network (*e.g.* fully connected layers, nonlinearities, *etc.*).
- Optionally: If we have additional meta-information $z$, then the above mentioned networks could be conditioned to obtain the conditioning mapping $\phi(x_m|z)$.

In other words, the key is to add up all representations and then apply nonlinear transformations.

**Equivariant model.** Our goal is to design neural network layers that are equivariant to the permutations of elements in the input $\mathbf{x}$. Based on Lemma 3, a neural network layer $\mathbf{f}_\Theta(\mathbf{x})$ is permutation equivariant if and only if all the off-diagonal elements of $\Theta$ are tied together and all the diagonal elements are equal as well, *i.e.*, $\Theta = \lambda \mathbf{I} + \gamma (\mathbf{1}\mathbf{1}^\mathsf{T})$ for $\lambda, \gamma \in \mathbb{R}$. This function is simply a non-linearity applied to a weighted combination of (i) its input $\mathbf{I}\mathbf{x}$ and; (ii) the sum of input values $(\mathbf{1}\mathbf{1}^\mathsf{T})\mathbf{x}$. Since summation does not depend on the permutation, the layer is permutation-equivariant. We can further manipulate the operations and parameters in this layer to get other **variations**, *e.g.*:

$$\mathbf{f}(\mathbf{x}) \doteq \boldsymbol{\sigma} \left( \lambda \mathbf{I}\mathbf{x} + \gamma \, \mathrm{maxpool}(\mathbf{x})\mathbf{1} \right). \tag{4}$$

where the maxpooling operation over elements of the set (similar to sum) is commutative. In practice, this variation performs better in some applications. This may be due to the fact that for $\lambda = \gamma$, the input to the non-linearity is max-normalized. Since composition of permutation equivariant functions is also permutation equivariant, we can build DeepSets by stacking such layers.

## 3.2 Other Related Works

Several recent works study equivariance and invariance in deep networks w.r.t. general group of transformations [12–14]. For example, [15] construct deep permutation invariant features by pairwise coupling of features at the previous layer, where $f_{i,j}([x_i, x_j]) \doteq [|x_i - x_j|, x_i + x_j]$ is invariant to transposition of $i$ and $j$. Pairwise interactions within sets have also been studied in [16, 17]. [18] approach unordered instances by finding "good" orderings.

The idea of pooling a function across set-members is not new. In [19], pooling was used binary classification task for causality on a set of samples. [20] use pooling across a panoramic projection of 3D object for classification, while [21] perform pooling across multiple views. [22] observe the invariance of the payoff matrix in normal form games to the permutation of its rows and columns (*i.e.* player actions) and leverage pooling to predict the player action. The need of permutation equivariance also arise in deep learning over sensor networks and multi-agent settings, where a special case of Lemma 3 has been used as the architecture [23].

In light of these related works, we would like to emphasize our novel contributions: (i) the universality result of Theorem 2 for permutation invariance that also relates DeepSets to other machine learning techniques, see Sec. 3; (ii) the permutation equivariant layer of (4), which, according to Lemma 3 identifies necessary and sufficient form of parameter-sharing in a standard neural layer and; (iii) novel application settings that we study next.

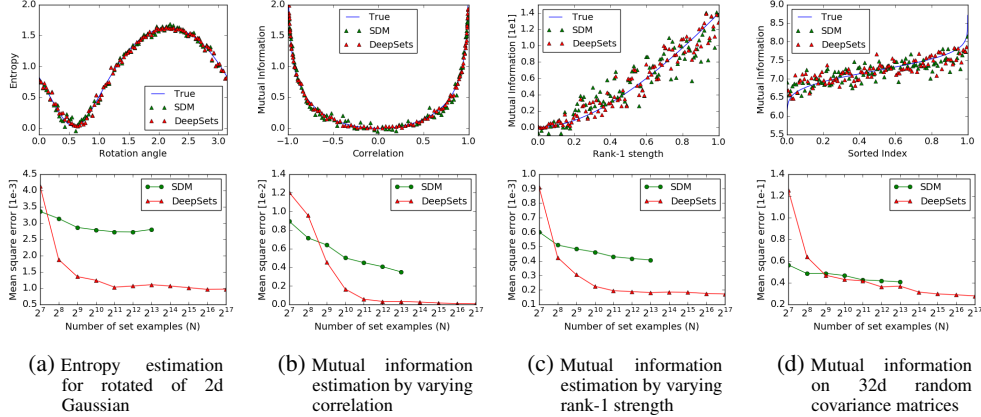

(a) Entropy estimation for rotated of 2d Gaussian

(b) Mutual information estimation by varying correlation

(c) Mutual information estimation by varying rank-1 strength

(d) Mutual information on 32d random covariance matrices

Figure 1: Population statistic estimation: Top set of figures, show prediction of DeepSets vs SDM for $N = 2^{10}$ case. Bottom set of figures, depict the mean squared error behavior as number of sets is increased. SDM has lower error for small $N$ and DeepSets requires more data to reach similar accuracy. But for high dimensional problems DeepSets easily *scales* to large number of examples and produces much *lower* estimation error. Note that the $N \times N$ matrix inversion in SDM makes it prohibitively expensive for $N > 2^{14} = 16384$.

# 4 Applications and Empirical Results

We present a diverse set of applications for DeepSets. For the supervised setting, we apply DeepSets to estimation of population statistics, sum of digits and classification of point-clouds, and regression with clustering side-information. The permutation-equivariant variation of DeepSets is applied to the task of outlier detection. Finally, we investigate the application of DeepSets to unsupervised set-expansion, in particular, concept-set retrieval and image tagging. In most cases we compare our approach with the state-of-the art and report competitive results.

## 4.1 Set Input Scalar Response

### 4.1.1 Supervised Learning: Learning to Estimate Population Statistics

In the first experiment, we learn entropy and mutual information of Gaussian distributions, without providing any information about Gaussianity to DeepSets. The Gaussians are generated as follows:

- Rotation: We randomly chose a $2 \times 2$ covariance matrix $\Sigma$, and then generated $N$ sample sets from $\mathcal{N}(0, R(\alpha)\Sigma R(\alpha)^T)$ of size $M = [300 - 500]$ for $N$ random values of $\alpha \in [0, \pi]$. Our goal was to learn the entropy of the marginal distribution of first dimension. $R(\alpha)$ is the rotation matrix.
- Correlation: We randomly chose a $d \times d$ covariance matrix $\Sigma$ for $d = 16$, and then generated $N$ sample sets from $\mathcal{N}(0, [\Sigma, \alpha\Sigma; \alpha\Sigma, \Sigma])$ of size $M = [300 - 500]$ for $N$ random values of $\alpha \in (-1, 1)$. Goal was to learn the mutual information of among the first $d$ and last $d$ dimension.
- Rank 1: We randomly chose $v \in \mathbb{R}^{32}$ and then generated a sample sets from $\mathcal{N}(0, I + \lambda vv^T)$ of size $M = [300 - 500]$ for $N$ random values of $\lambda \in (0, 1)$. Goal was to learn the mutual information.
- Random: We chose $N$ random $d \times d$ covariance matrices $\Sigma$ for $d = 32$, and using each, generated a sample set from $\mathcal{N}(0, \Sigma)$ of size $M = [300 - 500]$. Goal was to learn the mutual information.

We train using $L_2$ loss with a DeepSets architecture having 3 fully connected layers with ReLU activation for both transformations $\phi$ and $\rho$. We compare against Support Distribution Machines (SDM) using a RBF kernel [10], and analyze the results in Fig. 1.

### 4.1.2 Sum of Digits

Next, we compare to what happens if our set data is treated as a sequence. We consider the task of finding sum of a given set of digits. We consider two variants of this experiment:

**Text.** We randomly sample a subset of maximum $M = 10$ digits from this dataset to build $100k$ "sets" of training images, where the set-label is sum of digits in that set. We test against sums of $M$ digits, for $M$ starting from 5 all the way up to 100 over another $100k$ examples.

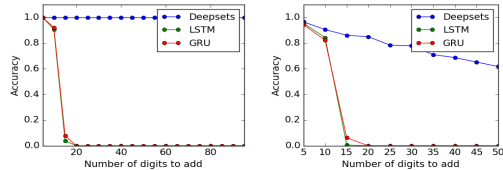

Figure 2: Accuracy of digit summation with text (*left*) and image (*right*) inputs. All approaches are trained on tasks of length 10 at most, tested on examples of length up to 100. We see that DeepSets generalizes better.

**Image.** MNIST8m [24] contains 8 million instances of $28 \times 28$ grey-scale stamps of digits in $\{0, \ldots, 9\}$. We randomly sample a subset of maximum $M = 10$ images from this dataset to build $N = 100k$ "sets" of training and $100k$ sets of test images, where the set-label is the sum of digits in that set (*i.e.* individual labels per image is unavailable). We test against sums of $M$ images of MNIST digits, for $M$ starting from 5 all the way up to 50.

We compare against recurrent neural networks – LSTM and GRU. All models are defined to have similar number of layers and parameters. The output of all models is a scalar, predicting the sum of $N$ digits. Training is done on tasks of length 10 at most, while at test time we use examples of length up to 100. The accuracy, *i.e.* exact equality after rounding, is shown in Fig. 2. DeepSets generalize much better. Note for image case, the best classification error for single digit is around $p = 0.01$ for MNIST8m, so in a collection of $N$ of images at least one image will be misclassified is $1 - (1 - p)^N$, which is 40% for $N = 50$. This matches closely with observed value in Fig. 2(b).

### 4.1.3 Point Cloud Classification

A point-cloud is a set of low-dimensional vectors. This type of data is frequently encountered in various applications like robotics, vision, and cosmology. In these applications, existing methods often convert the point-cloud data to voxel or mesh representation as a preprocessing step, *e.g.* [26, 29, 30]. Since the output of many range sensors, such as LiDAR, is in the form of point-cloud, direct application of deep learning methods to point-cloud is highly desirable. Moreover, it is easy and cheaper to apply transformations, such as rotation and translation, when working with point-clouds than voxelized 3D objects.

As point-cloud data is just a set of points, we can use DeepSets to classify point-cloud representation of a subset of ShapeNet objects [31], called ModelNet40 [25]. This subset consists of 3D representation of 9,843 training and 2,468

| Model | Instance Size | Representation | Accuracy |
|---|---|---|---|
| 3DShapeNets [25] | $30^3$ | voxels (using convolutional deep belief net) | 77% |
| VoxNet [26] | $32^3$ | voxels (voxels from point-cloud + 3D CNN) | 83.10% |
| MVCNN [21] | $164 \times 164 \times 12$ | multi-vew images (2D CNN + view-pooling) | 90.1% |
| VRN Ensemble [27] | $32^3$ | voxels (3D CNN, variational autoencoder) | 95.54% |
| 3D GAN [28] | $64^3$ | voxels (3D CNN, generative adversarial training) | 83.3% |
| DeepSets | **$5000 \times 3$** | point-cloud | $90 \pm .3\%$ |
| DeepSets | **$100 \times 3$** | point-cloud | $82 \pm 2\%$ |

Table 1: Classification accuracy and the representation-size used by different methods on the ModelNet40.

test instances belonging to 40 classes of objects. We produce point-clouds with 100, 1000 and 5000 particles each ($x, y, z$-coordinates) from the mesh representation of objects using the point-cloud-library's sampling routine [32]. Each set is normalized by the initial layer of the deep network to have zero mean (along individual axes) and unit (global) variance. Tab. 1 compares our method using three permutation equivariant layers against the competition; see Appendix H for details.

### 4.1.4 Improved Red-shift Estimation Using Clustering Information

An important regression problem in cosmology is to estimate the red-shift of galaxies, corresponding to their age as well as their distance from us [33] based on photometric observations. One way to estimate the red-shift from photometric observations is using a regression model [34] on the galaxy clusters. The prediction for each galaxy does not change by permuting the members of the galaxy cluster. Therefore, we can treat each galaxy cluster as a "set" and use DeepSets to estimate the individual galaxy red-shifts. See Appendix G for more details.

For each galaxy, we have 17 photometric features from the redMaPPer galaxy cluster catalog [35] that contains photometric readings for 26,111 red galaxy clusters. Each galaxy-cluster in this catalog has between $\sim 20 - 300$ galaxies – *i.e.* $\mathbf{x} \in \mathbb{R}^{N(c) \times 17}$, where $N(c)$ is the cluster-size. The catalog also provides accurate spectroscopic red-shift estimates for a *subset* of these galaxies.

| Method | Scatter |
|---|---|
| MLP | 0.026 |
| redMaPPer | 0.025 |
| DeepSets | 0.023 |

Table 2: Red-shift experiment. Lower scatter is better.

We randomly split the data into 90% training and 10% test clusters, and minimize the squared loss of the prediction for available spectroscopic red-shifts. As it is customary in cosmology literature, we report the average **scatter** $\frac{|z_{\text{spec}} - z|}{1 + z_{\text{spec}}}$, where $z_{\text{spec}}$ is the accurate spectroscopic measurement and $z$ is a photometric estimate in Tab. 2.

| Method | LDA-1$k$ (Vocab = 17$k$) | | | | | LDA-3$k$ (Vocab = 38$k$) | | | | | LDA-5$k$ (Vocab = 61$k$) | | | | |
|---|---|---|---|---|---|---|---|---|---|---|---|---|---|---|---|
| | Recall (%) | | | MRR | Med. | Recall (%) | | | MRR | Med. | Recall (%) | | | MRR | Med. |
| | @10 | @100 | @1k | | | @10 | @100 | @1k | | | @10 | @100 | @1k | | |
| Random | 0.06 | 0.6 | 5.9 | 0.001 | 8520 | 0.02 | 0.2 | 2.6 | 0.000 | 28635 | 0.01 | 0.2 | 1.6 | 0.000 | 30600 |
| Bayes Set | 1.69 | 11.9 | 37.2 | 0.007 | 2848 | 2.01 | 14.5 | 36.5 | 0.008 | 3234 | 1.75 | 12.5 | 34.5 | 0.007 | 3590 |
| w2v Near | **6.00** | **28.1** | **54.7** | 0.021 | **641** | 4.80 | 21.2 | 43.2 | 0.016 | 2054 | 4.03 | 16.7 | 35.2 | 0.013 | 6900 |
| NN-max | 4.78 | 22.5 | 53.1 | 0.023 | 779 | 5.30 | 24.9 | 54.8 | 0.025 | 672 | 4.72 | 21.4 | 47.0 | 0.022 | 1320 |
| NN-sum-con | 4.58 | 19.8 | 48.5 | 0.021 | 1110 | 5.81 | 27.2 | 60.0 | **0.027** | 453 | 4.87 | 23.5 | 53.9 | 0.022 | 731 |
| NN-max-con | 3.36 | 16.9 | 46.6 | 0.018 | 1250 | 5.61 | 25.7 | 57.5 | 0.026 | 570 | 4.72 | 22.0 | 51.8 | 0.022 | 877 |
| DeepSets | 5.53 | 24.2 | 54.3 | **0.025** | 696 | **6.04** | **28.5** | **60.7** | **0.027** | **426** | **5.54** | **26.1** | **55.5** | **0.026** | **616** |

Table 3: Results on Text Concept Set Retrieval on LDA-1k, LDA-3k, and LDA-5k. Our DeepSets model outperforms other methods on LDA-3k and LDA-5k. However, all neural network based methods have inferior performance to w2v-Near baseline on LDA-1k, possibly due to small data size. Higher the better for recall@k and mean reciprocal rank (MRR). Lower the better for median rank (Med.)

## 4.2 Set Expansion

In the set expansion task, we are given a set of objects that are similar to each other and our goal is to find new objects from a large pool of candidates such that the selected new objects are similar to the query set. To achieve this one needs to reason out the concept connecting the given set and then retrieve words based on their relevance to the inferred concept. It is an important task due to wide range of potential applications including personalized information retrieval, computational advertisement, tagging large amounts of unlabeled or weakly labeled datasets.

Going back to de Finetti's theorem in Sec. 3.2, where we consider the marginal probability of a set of observations, the marginal probability allows for very simple metric for scoring additional elements to be added to $X$. In other words, this allows one to perform set expansion via the following score

$$s(x|X) = \log p(X \cup \{x\} \,|\alpha) - \log p(X|\alpha)p(\{x\} \,|\alpha) \qquad (5)$$

Note that $s(x|X)$ is the point-wise mutual information between $x$ and $X$. Moreover, due to exchangeability, it follows that regardless of the order of elements we have

$$S(X) = \sum_m s\left(x_m| \{x_{m-1}, \dots x_1\}\right) = \log p(X|\alpha) - \sum_{m=1}^{M} \log p(\{x_m\} \,|\alpha) \qquad (6)$$

When inferring sets, our goal is to find set completions $\{x_{m+1}, \dots x_M\}$ for an initial set of query terms $\{x_1, \dots, x_m\}$, such that the aggregate set is coherent. This is the key idea of the Bayesian Set algorithm [36] (details in Appendix D). Using DeepSets, we can solve this problem in more generality as we can drop the assumption of data belonging to certain exponential family.

For learning the score $s(x|X)$, we take recourse to large-margin classification with structured loss functions [37] to obtain the relative loss objective $l(x, x'|X) = \max(0, s(x'|X) - s(x|X) + \Delta(x, x'))$. In other words, we want to ensure that $s(x|X) \geq s(x'|X) + \Delta(x, x')$ whenever $x$ should be added and $x'$ should not be added to $X$.

**Conditioning.** Often machine learning problems do not exist in isolation. For example, task like tag completion from a given set of tags is usually related to an object $z$, for example an image, that needs to be tagged. Such meta-data are usually abundant, *e.g.* author information in case of text, contextual data such as the user click history, or extra information collected with LiDAR point cloud.

Conditioning graphical models with meta-data is often complicated. For instance, in the Beta-Binomial model we need to ensure that the counts are always nonnegative, regardless of $z$. Fortunately, DeepSets does not suffer from such complications and the fusion of multiple sources of data can be done in a relatively straightforward manner. Any of the existing methods in deep learning, including feature concatenation by averaging, or by max-pooling, can be employed. Incorporating these meta-data often leads to significantly improved performance as will be shown in experiments; Sec. 4.2.2.

### 4.2.1 Text Concept Set Retrieval

In text concept set retrieval, the objective is to retrieve words belonging to a 'concept' or 'cluster', given few words from that particular concept. For example, given the set of words {*tiger*, *lion*, *cheetah*}, we would need to retrieve other related words like *jaguar*, *puma*, *etc*, which belong to the same concept of big cats. This task of concept set retrieval can be seen as a set completion task conditioned on the latent semantic concept, and therefore our DeepSets form a desirable approach.

**Dataset.** We construct a large dataset containing sets of $N_T = 50$ related words by extracting topics from latent Dirichlet allocation [38, 39], taken out-of-the-box[1]. To compare across scales, we

consider three values of $k = \{1k, 3k, 5k\}$ giving us three datasets LDA-$1k$, LDA-$3k$, and LDA-$5k$, with corresponding vocabulary sizes of $17k, 38k$, and $61k$.

**Methods.** We learn this using a margin loss with a DeepSets architecture having 3 fully connected layers with ReLU activation for both transformations $\phi$ and $\rho$. Details of the architecture and training are in Appendix E. We compare to several baselines: (a) **Random** picks a word from the vocabulary uniformly at random. (b) **Bayes Set** [36]. (c) **w2v-Near** computes the nearest neighbors in the word2vec [40] space. Note that both Bayes Set and w2v NN are strong baselines. The former runs Bayesian inference using Beta-Binomial conjugate pair, while the latter uses the powerful 300 dimensional word2vec trained on the billion word GoogleNews corpus[2]. (d) **NN-max** uses a similar architecture as our DeepSets but uses max pooling to compute the set feature, as opposed to sum pooling. (e) **NN-max-con** uses max pooling on set elements but concatenates this pooled representation with that of query for a final set feature. (f) **NN-sum-con** is similar to NN-max-con but uses sum pooling followed by concatenation with query representation.

**Evaluation.** We consider the standard retrieval metrics – recall@K, median rank and mean reciprocal rank, for evaluation. To elaborate, recall@K measures the number of true labels that were recovered in the top K retrieved words. We use three values of K = $\{10, 100, 1k\}$. The other two metrics, as the names suggest, are the median and mean of reciprocals of the true label ranks, respectively. Each dataset is split into TRAIN (80%), VAL (10%) and TEST (10%). We learn models using TRAIN and evaluate on TEST, while VAL is used for hyperparameter selection and early stopping.

**Results and Observations.** As seen in Tab. 3: (a) Our DeepSets model outperforms all other approaches on LDA-$3k$ and LDA-$5k$ by any metric, highlighting the significance of permutation invariance property. (b) On LDA-$1k$, our model does not perform well when compared to w2v-Near. We hypothesize that this is due to small size of the dataset insufficient to train a high capacity neural network, while w2v-Near has been trained on a billion word corpus. Nevertheless, our approach comes the closest to w2v-Near amongst other approaches, and is only 0.5% lower by Recall@10.

### 4.2.2 Image Tagging

We next experiment with image tagging, where the task is to retrieve all relevant tags corresponding to an image. Images usually have only a subset of relevant tags, therefore predicting other tags can help enrich information that can further be leveraged in a downstream supervised task. In our setup, we learn to predict tags by conditioning DeepSets on the image, *i.e.*, we train to predict a partial set of tags from the image and remaining tags. At test time, we predict tags from the image alone.

| Method | ESP game | | | | IAPRTC-12.5 | | | |
|---|---|---|---|---|---|---|---|---|
| | P | R | F1 | N+ | P | R | F1 | N+ |
| Least Sq. | 35 | 19 | 25 | 215 | 40 | 19 | 26 | 198 |
| MBRM | 18 | 19 | 18 | 209 | 24 | 23 | 23 | 223 |
| JEC | 24 | 19 | 21 | 222 | 29 | 19 | 23 | 211 |
| FastTag | 46 | 22 | 30 | **247** | 47 | 26 | 34 | **280** |
| Least Sq.(D) | **44** | 32 | **37** | 232 | 46 | 30 | 36 | 218 |
| FastTag(D) | **44** | 32 | **37** | 229 | 46 | **33** | **38** | 254 |
| DeepSets | 39 | **34** | 36 | 246 | 42 | 31 | 36 | 247 |

Table 4: Results of image tagging on ESPgame and IAPRTC-12.5 datasets. Performance of our DeepSets approach is roughly similar to the best competing approaches, except for precision. Refer text for more details. Higher the better for all metrics – precision (P), recall (R), f1 score (F1), and number of non-zero recall tags (N+).

**Datasets.** We report results on the following three datasets - ESPGame, IAPRTC-12.5 and our in-house dataset, COCO-Tag. We refer the reader to Appendix F, for more details about datasets.

**Methods.** The setup for DeepSets to tag images is similar to that described in Sec. 4.2.1. The only difference being the conditioning on the image features, which is concatenated with the set feature obtained from pooling individual element representations.

**Baselines.** We perform comparisons against several baselines, previously reported in [41]. Specifically, we have Least Sq., a ridge regression model, MBRM [42], JEC [43] and FastTag [41]. Note that these methods do not use deep features for images, which could lead to an unfair comparison. As there is no publicly available code for MBRM and JEC, we cannot get performances of these models with Resnet extracted features. However, we report results with deep features for FastTag and Least Sq., using code made available by the authors[3].

**Evaluation.** For ESPgame and IAPRTC-12.5, we follow the evaluation metrics as in [44]–precision (P), recall (R), F1 score (F1), and number of tags with non-zero recall (N+). These metrics are evaluate for each tag and the mean is reported (see [44] for further details). For COCO-Tag, however, we use recall@K for three values of K = $\{10, 100, 1000\}$, along with median rank and mean reciprocal rank (see evaluation in Sec. 4.2.1 for metric details).

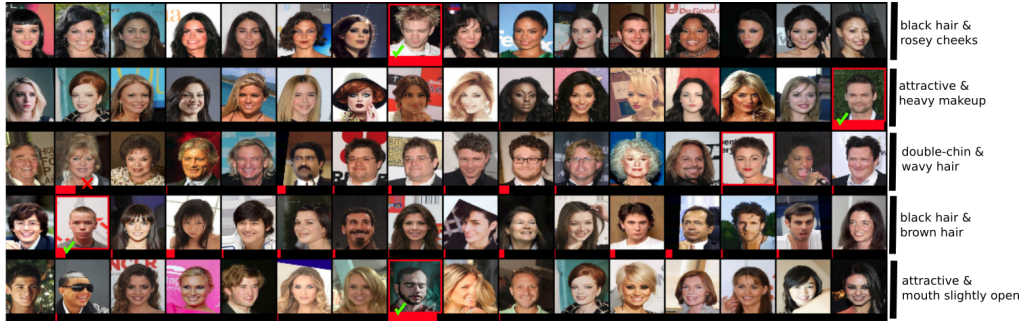

Figure 3: Each row shows a set, constructed from CelebA dataset, such that all set members except for an outlier, share at least two attributes (on the right). The **outlier is identified with a red frame**. The model is trained by observing examples of sets and their anomalous members, **without access to the attributes**. The probability assigned to each member by the outlier detection network is visualized using a **red bar** at the bottom of each image. The probabilities in each row sum to one.

**Results and Observations.** Tab. 4 shows results of image tagging on ESPgame and IAPRTC-12.5, and Tab. 5 on COCO-Tag. Here are the key observations from Tab. 4: (a) performance of our DeepSets model is comparable to the best approaches on all metrics but precision, (b) our recall beats the best approach by 2% in ESPgame. On further investigation, we found that the DeepSets model retrieves more relevant tags, which are not present in list of ground truth tags due to a limited 5 tag annotation. Thus, this takes a toll on precision while gaining on recall, yet

| Method | Recall | | | MRR | Med. |
|---|---|---|---|---|---|
| | @10 | @100 | @1k | | |
| w2v NN (blind) | 5.6 | 20.0 | 54.2 | 0.021 | 823 |
| DeepSets (blind) | 9.0 | 39.2 | 71.3 | 0.044 | 310 |
| DeepSets | **31.4** | **73.4** | **95.3** | **0.131** | **28** |

Table 5: Results on COCO-Tag dataset. Clearly, DeepSets outperforms other baselines significantly. Higher the better for recall@K and mean reciprocal rank (MRR). Lower the better for median rank (Med).

yielding improvement on F1. On the larger and richer COCO-Tag, we see that the DeepSets approach outperforms other methods comprehensively, as expected. Qualitative examples are in Appendix F.

## 4.3 Set Anomaly Detection

The objective here is to find the anomalous face in each set, simply by observing examples and without any access to the attribute values. CelebA dataset [45] contains 202,599 face images, each annotated with 40 boolean attributes. We build $N = 18,000$ sets of $64 \times 64$ stamps, using these attributes each containing $M = 16$ images (on the training set) as follows: randomly select 2 attributes, draw 15 images having those attributes, and a single target image where both attributes are absent. Using a similar procedure we build sets on the test images. No individual person's face appears in both train and test sets. Our deep neural network consists of 9 2D-convolution and max-pooling layers followed by 3 permutation-equivariant layers, and finally a softmax layer that assigns a probability value to each set member (Note that one could identify arbitrary number of outliers using a sigmoid activation at the output). Our trained model successfully finds the anomalous face in **75% of test sets**. Visually inspecting these instances suggests that the task is non-trivial even for humans; see Fig. 3.

As a *baseline*, we repeat the same experiment by using a set-pooling layer after convolution layers, and replacing the permutation-equivariant layers with fully connected layers of same size, where the final layer is a 16-way softmax. The resulting network shares the convolution filters for all instances within all sets, however the input to the softmax is not equivariant to the permutation of input images. Permutation equivariance seems to be crucial here as the baseline model achieves a training and **test accuracy of** $\sim 6.3\%$; the same as random selection. See Appendix I for more details.

## 5 Summary

In this paper, we develop DeepSets, a model based on powerful permutation invariance and equivariance properties, along with the theory to support its performance. We demonstrate the generalization ability of DeepSets across several domains by extensive experiments, and show both qualitative and quantitative results. In particular, we explicitly show that DeepSets outperforms other intuitive deep networks, which are not backed by theory (Sec. 4.2.1, Sec. 4.1.2). Last but not least, it is worth noting that the state-of-the-art we compare to is a specialized technique for each task, whereas our one model, *i.e.*, DeepSets, is competitive across the board.

## Footnotes

[1] `github.com/dmlc/experimental-lda`

[2]`code.google.com/archive/p/word2vec/`

[3]`http://www.cse.wustl.edu/~mchen/`

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
