[Supplementary Material]

# Appendix: Deep Sets

## A  Proofs and Discussion Related to Theorem 2

A function $f$ transforms its domain $\mathcal{X}$ into its range $\mathcal{Y}$. Usually, the input domain is a vector space $\mathbb{R}^d$ and the output response range is either a discrete space, e.g. $\{0, 1\}$ in case of classification, or a continuous space $\mathbb{R}$ in case of regression.

Now, if the input is a set $X = \{x_1, \ldots, x_M\}, x_m \in \mathfrak{X}$, i.e. $\mathcal{X} = 2^{\mathfrak{X}}$, then we would like the response of the function not to depend on the ordering of the elements in the set. In other words,

**Property 1**  *A function $f : 2^{\mathfrak{X}} \to \mathbb{R}$ acting on sets must be permutation invariant to the order of objects in the set, i.e.*

$$f(\{x_1, ..., x_M\}) = f(\{x_{\pi(1)}, ..., x_{\pi(M)}\}) \tag{7}$$

*for any permutation $\pi$.*

Now, roughly speaking, we claim that such functions must have a structure of the form $f(X) = \rho\left(\sum_{x \in X} \phi(x)\right)$ for some functions $\rho$ and $\phi$. Over the next two sections we try to formally prove this structure of the permutation invariant functions.

### A.1  Countable Case

**Theorem 2**  *Assume the elements are countable, i.e. $|\mathfrak{X}| < \aleph_0$. A function $f : 2^{\mathfrak{X}} \to \mathbb{R}$ operating on a set $X$ can be a valid set function, i.e. it is permutation invariant to the elements in $X$, if and only if it can be decomposed in the form $\rho\left(\sum_{x \in X} \phi(x)\right)$, for suitable transformations $\phi$ and $\rho$.*

**Proof.** Permutation invariance follows from the fact that sets have no particular order, hence any function on a set must not exploit any particular order either. The sufficiency follows by observing that the function $\rho\left(\sum_{x \in X} \phi(x)\right)$ satisfies the permutation invariance condition.

To prove necessity, i.e. that all functions can be decomposed in this manner, we begin by noting that there must be a mapping from the elements to natural numbers functions, since the elements are countable. Let this mapping be denoted by $c : \mathfrak{X} \to \mathbb{N}$. Now if we let $\phi(x) = 4^{-c(x)}$ then $\sum_{x \in X} \phi(x)$ constitutes an unique representation for every set $X \in 2^{\mathfrak{X}}$. Now a function $\rho : \mathbb{R} \to \mathbb{R}$ can always be constructed such that $f(X) = \rho\left(\sum_{x \in X} \phi(x)\right)$. ∎

### A.2  Uncountable Case

The extension to case when $\mathfrak{X}$ is uncountable, *e.g.* $\mathfrak{X} = [0, 1]$, is not so trivial. We could only prove in case of fixed set size, *e.g.* $\mathcal{X} = [0, 1]^M$ instead of $\mathcal{X} = 2^{\mathfrak{X}} = 2^{[0,1]}$, that any permutation invariant continuous function can be expressed as $\rho\left(\sum_{x \in X} \phi(x)\right)$. Also, we show that there is a universal approximator of the same form. These results are discussed below.

To illustrate the uncountable case, we assume a fixed set size of $M$. Without loss of generality we can let $\mathfrak{X} = [0, 1]$. Then the domain becomes $[0, 1]^M$. Also, to handle ambiguity due to permutation, we often define the domain to be the set $\mathcal{X} = \{(x_1, ..., x_M) \in [0, 1]^M : x_1 \leq x_2 \leq \cdots \leq x_M\}$ for some ordering of the elements in $\mathfrak{X}$.

The proof builds on the famous Newton-Girard formulae which connect moments of a sample set (sum-of-power) to the elementary symmetric polynomials. But first we present some results needed for the proof. The first result establishes that sum-of-power mapping is injective.

**Lemma 4** *Let $\mathcal{X} = \{(x_1, ..., x_M) \in [0, 1]^M : x_1 \leq x_2 \leq \cdots \leq x_M\}$. The sum-of-power mapping $E : \mathcal{X} \to \mathbb{R}^{M+1}$ defined by the coordinate functions*

$$Z_q := E_q(X) := \sum_{m=1}^{M} (x_m)^q, \qquad q = 0, ..., M. \tag{8}$$

*is injective.*

**Proof.** Suppose for some $u, v \in \mathcal{X}$, we have $E(u) = E(v)$. We will now show that it must be the case that $u = v$. Construct two polynomials as follows:

$$P_u(x) = \prod_{m=1}^{M} (x - u_m) \qquad P_v(x) = \prod_{m=1}^{M} (x - v_m) \tag{9}$$

If we expand the two polynomials we obtain:

$$P_u(x) = x^M - a_1 x^{M-1} + \cdots (-1)^{M-1} a_{M-1} x + (-1)^M a_M$$
$$P_v(x) = x^M - b_1 x^{M-1} + \cdots (-1)^{M-1} b_{M-1} x + (-1)^M b_M \tag{10}$$

with coefficients being elementary symmetric polynomials in $u$ and $v$ respectively, *i.e.*

$$a_m = \sum_{1 \leq j_1 < j_2 < \cdots < j_m \leq M} u_{j_1} u_{j_2} \cdots u_{j_m} \qquad b_m = \sum_{1 \leq j_1 < j_2 < \cdots < j_m \leq M} v_{j_1} v_{j_2} \cdots v_{j_m} \tag{11}$$

These elementary symmetric polynomials can be uniquely expressed as a function of $E(u)$ and $E(v)$ respectively, by Newton-Girard formula. The $m$-th coefficient is given by the determinant of $m \times m$ matrix having terms from $E(u)$ and $E(v)$ respectively:

$$a_m = \frac{1}{m} \det \begin{pmatrix} E_1(u) & 1 & 0 & 0 & \cdots & 0 \\ E_2(u) & E_1(u) & 1 & 0 & \cdots & 0 \\ E_3(u) & E_2(u) & E_1(u) & 1 & \cdots & 0 \\ \vdots & \vdots & \vdots & \vdots & \ddots & \vdots \\ E_{m-1}(u) & E_{m-2}(u) & E_{m-3}(u) & E_{m-4}(u) & \cdots & 1 \\ E_m(u) & E_{m-1}(u) & E_{m-2}(u) & E_{m-3}(u) & \cdots & E_1(u) \end{pmatrix}$$

$$\tag{12}$$

$$b_m = \frac{1}{m} \det \begin{pmatrix} E_1(v) & 1 & 0 & 0 & \cdots & 0 \\ E_2(v) & E_1(v) & 1 & 0 & \cdots & 0 \\ E_3(v) & E_2(v) & E_1(v) & 1 & \cdots & 0 \\ \vdots & \vdots & \vdots & \vdots & \ddots & \vdots \\ E_{m-1}(v) & E_{m-2}(v) & E_{m-3}(v) & E_{m-4}(v) & \cdots & 1 \\ E_m(v) & E_{m-1}(v) & E_{m-2}(v) & E_{m-3}(v) & \cdots & E_1(v) \end{pmatrix}$$

Since we assumed $E(u) = E(v)$ implying $[a_1, ..., a_M] = [b_1, ..., b_M]$, which in turn implies that the polynomials $P_u$ and $P_v$ are the same. Therefore, their roots must be the same, which shows that $u = v$. ∎

The second result we borrow from [46] which establishes a homeomorphism between coefficients and roots of a polynomial.

**Theorem 5 [46]** *The function* $f : \mathbb{C}^M \to \mathbb{C}^M$, *which associates every* $a \in \mathbb{C}^M$ *to the multiset of roots,* $f(a) \in \mathbb{C}^M$, *of the monic polynomial formed using* $a$ *as the coefficient* i.e. $x^M + a_1 x^{M-1} + \cdots (-1)^{M-1} a_{M-1} x + (-1)^M a_M$, *is a homeomorphism.*

Among other things, this implies that (complex) roots of a polynomial depends continuously on the coefficients. We will use this fact for our next lemma.

Finally, we establish a continuous inverse mapping for the sum-of-power function.

**Lemma 6** *Let* $\mathcal{X} = \{(x_1, ..., x_M) \in [0, 1]^M : x_1 \leq x_2 \leq \cdots \leq x_M\}$. *We define the sum-of-power mapping* $E : \mathcal{X} \to \mathcal{Z}$ *by the coordinate functions*

$$Z_q := E_q(X) := \sum_{m=1}^{M} (x_m)^q, \qquad q = 0, ..., M. \tag{13}$$

*where* $\mathcal{Z}$ *is the range of the function. The function* $E$ *has a continuous inverse mapping.*

**Proof.** First of all note that $\mathcal{Z}$, the range of $E$, is a compact set. This follows from following observations:

- The domain of $E$ is a bounded polytope (*i.e.* a compact set),
- $E$ is a continuous function, and
- image of a compact set under a continuous function is a compact set.

To show the continuity of inverse mapping, we establish connection to the continuous dependence of roots of polynomials on its coefficients.

As in Lemma 4, for any $u \in \mathcal{X}$, let $z = E(u)$ and construct the polynomial:

$$P_u(x) = \prod_{m=1}^{M} (x - u_m) \tag{14}$$

If we expand the polynomial we obtain:

$$P_u(x) = x^M - a_1 x^{M-1} + \cdots (-1)^{M-1} a_{M-1} x + (-1)^M a_M \tag{15}$$

with coefficients being elementary symmetric polynomials in $u$, *i.e.*

$$a_m = \sum_{1 \leq j_1 < j_2 < \cdots < j_m \leq M} u_{j_1} u_{j_2} \cdots u_{j_m} \tag{16}$$

These elementary symmetric polynomials can be uniquely expressed as a function of $z$ by Newton-Girard formula:

$$a_m = \frac{1}{m} \det \begin{pmatrix} z_1 & 1 & 0 & 0 & \cdots & 0 \\ z_2 & z_1 & 1 & 0 & \cdots & 0 \\ z_3 & z_2 & z_1 & 1 & \cdots & 0 \\ \vdots & \vdots & \vdots & \vdots & \ddots & \vdots \\ z_{m-1} & z_{m-2} & z_{m-3} & z_{m-4} & \cdots & 1 \\ z_m & z_{m-1} & z_{m-2} & z_{m-3} & \cdots & z_1 \end{pmatrix} \tag{17}$$

Since determinants are just polynomials, $a$ is a continuous function of $z$. Thus to show continuity of inverse mapping of $E$, it remains to show continuity from $a$ back to the roots $u$. In this regard, we invoke Theorem 5. Note that homeomorphism implies the mapping as well as its inverse is continuous. Thus, restricting to the compact set $\mathcal{Z}$ where the map from coefficients to roots only goes to the reals, the desired result follows. To explicitly check the continuity, note that limit of $E^{-1}(z)$, as $z$ approaches $z^*$ from inside $\mathcal{Z}$, always exists and is equal to $E^{-1}(z^*)$ since it does so in the complex plane. ∎

With the lemma developed above we are in a position to tackle the main theorem.

**Theorem 7** *Let $f : [0,1]^M \to \mathbb{R}$ be a permutation invariant continuous function iff it has the representation*

$$f(x_1, ..., x_M) = \rho \left( \sum_{m=1}^{M} \phi(x_m) \right) \tag{18}$$

*for some continuous outer and inner function $\rho : \mathbb{R}^{M+1} \to \mathbb{R}$ and $\phi : \mathbb{R} \to \mathbb{R}^{M+1}$ respectively. The inner function $\phi$ is independent of the function $f$.*

**Proof.** The sufficiency follows by observing that the function $\rho \left( \sum_{m=1}^{M} \phi(x_m) \right)$ satisfies the permutation invariance condition.

To prove necessity, *i.e.* that all permutation invariant continuous functions over the compact set can be expressed in this manner, we divide the proof into two parts, with outline in Fig. 4. We begin

Figure 4: Outline of the proof strategy for Theorem 2.1. The proof consists of two parts. First, we desire to show that we can find unique embeddings for each possible input, *i.e.* we show that there exists a homeomorphism $E$ of the form $E(X) = \sum_{x \in X} \phi(x)$ between original domain and some higher dimensional space $\mathcal{Z}$. The second part of the proof consists of showing we can map the embedding to desired target value, *i.e.* to show the existence of the continuous map $\rho$ between $\mathcal{Z}$ and original target space such that $f(X) = \rho(\sum_{x \in X} \phi(x))$.

by looking at the continuous embedding formed by the inner function: $E(X) = \sum_{m=1}^{M} \phi(x_m)$. Consider $\phi : \mathbb{R} \to \mathbb{R}^{M+1}$ defined as $\phi(x) = [1, x, x^2, ..., x^M]$. Now as $E$ is a polynomial, the image of $[0, 1]^M$ in $\mathbb{R}^{M+1}$ under $E$ is a compact set as well, denote it by $\mathcal{Z}$. Then by definition, the embedding $E : [0, 1]^M \to \mathcal{Z}$ is surjective. Using Lemma 4 and 6, we know that upon restricting the permutations, *i.e.* replacing $[0, 1]^M$ with $\mathcal{X} = \{(x_1, ..., x_M) \in [0, 1]^M : x_1 \le x_2 \le \cdots \le x_M\}$, the embedding $E : \mathcal{X} \to \mathcal{Z}$ is injective with a continuous inverse. Therefore, combining these observation we get that $E$ is a homeomorphism between $\mathcal{X}$ and $\mathcal{Z}$. Now it remains to show that we can map the embedding to desired target value, *i.e.* to show the existence of the continuous map $\rho : \mathcal{Z} \to \mathbb{R}$ such that $\rho(E(X)) = f(X)$. In particular consider the map $\rho(z) = f(E^{-1}(z))$. The continuity of $\rho$ follows directly from the fact that composition of continuous functions is continuous. Therefore we can always find continuous functions $\phi$ and $\rho$ to express any permutation invariant function $f$ as $\rho \left( \sum_{m=1}^{M} \phi(x_m) \right)$. ∎

A very similar but more general results holds in case of any continuous function (not necessarily permutation invariant). The result is known as Kolmogorov-Arnold representation theorem [47, Chap. 17] which we state below:

**Theorem 8 (*Kolmogorov–Arnold representation*)** *Let $f : [0, 1]^M \to \mathbb{R}$ be an arbitrary multivariate continuous function iff it has the representation*

$$f(x_1, ..., x_M) = \rho \left( \sum_{m=1}^{M} \lambda_m \phi(x_m) \right) \tag{19}$$

*with continuous outer and inner functions $\rho : \mathbb{R}^{2M+1} \to \mathbb{R}$ and $\phi : \mathbb{R} \to \mathbb{R}^{2M+1}$. The inner function $\phi$ is independent of the function $f$.*

This theorem essentially states a representation theorem for any multivariate continuous function. Their representation is very similar to the one we are proved, except for the dependence of inner transformation on the co-ordinate through $\lambda_m$. Thus it is reassuring that behind all the beautiful mathematics something intuitive is happening. If the function is permutation invariant, this dependence on co-ordinate of the inner transformation gets dropped!

Further we can show that arbitrary approximator having the same form can be obtained for continuous permutation-invariant functions.

**Theorem 9** *Assume the elements are from a compact set in $\mathbb{R}^d$,* i.e. *possibly uncountable, and the set size is fixed to $M$. Then any continuous function operating on a set $X$, i.e. $f : \mathbb{R}^{d \times M} \to \mathbb{R}$ which is permutation invariant to the elements in $X$ can be approximated arbitrarily close in the form of $\rho \left( \sum_{x \in X} \phi(x) \right)$, for suitable transformations $\phi$ and $\rho$.*

**Proof.** Permutation invariance follows from the fact that sets have no particular order, hence any function on a set must not exploit any particular order either. The sufficiency follows by observing that the function $\rho \left( \sum_{x \in X} \phi(x) \right)$ satisfies the permutation invariance condition.

To prove necessity, *i.e.* that all continuous functions over the compact set can be approximated arbitrarily close in this manner, we begin noting that polynomials are universal approximators by Stone–Weierstrass theorem [48, sec. 5.7]. In this case the Chevalley-Shephard-Todd (CST) theorem [49, chap. V, theorem 4], or more precisely, its special case, the Fundamental Theorem of Symmetric Functions states that symmetric polynomials are given by a polynomial of homogeneous symmetric monomials. The latter are given by the sum over monomial terms, which is all that we need since it implies that all symmetric polynomials can be written in the form required by the theorem. ∎

Finally, we still conjecture that even in case of sets of all sizes, *i.e.* when the domain is $2^{[0,1]}$, a representation of the form $f(X) = \rho \left( \sum_{x \in X} \phi(x) \right)$ should exist for all "continuous" permutation invariant functions for some suitable transformations $\rho$ and $\phi$. However, in this case even what a "continuous" function means is not clear as the space $2^{[0,1]}$ does not have any natural topology. As a future work, we want to study further by defining various topologies, like using Fréchet distance as used in [46] or MMD distance. Our preliminary findings in this regards hints that using MMD distance if the representation is allowed to be in $\ell^2$, instead of being finite dimensional, then the conjecture seems to be provable. Thus, clearly this direction needs further exploration. We end this section by providing some examples:

**Examples:**

- $x_1 x_2 (x_1 + x_2 + 3)$, Consider $\phi(x) = [x, x^2, x^3]$ and $\rho([u, v, w]) = uv - w + 3(u^2 - v)/2$, then $\rho(\phi(x_1) + \phi(x_2))$ is the desired function.

- $x_1 x_2 x_3 + x_1 + x_2 + x_3$, Consider $\phi(x) = [x, x^2, x^3]$ and $\rho([u, v, w]) = (u^3 + 2w - 3uv)/6 + u$, then $\rho(\phi(x_1) + \phi(x_2) + \phi(x_3))$ is the desired function.

- $1/n(x_1 + x_2 + x_3 + ... + x_m)$, Consider $\phi(x) = [1, x]$ and $\rho([u, v]) = v/u$, then $\rho(\phi(x_1) + \phi(x_2) + \phi(x_3) + ... + \phi(x_m))$ is the desired function.

- $\max\{x_1, x_2, x_3, ..., x_m\}$, Consider $\phi(x) = [e^{\alpha x}, x e^{\alpha x}]$ and $\rho([u, v]) = v/u$, then as $\alpha \to \infty$, then we have $\rho(\phi(x_1) + \phi(x_2) + \phi(x_3) + ... + \phi(x_m))$ approaching the desired function.

- Second largest among $\{x_1, x_2, x_3, ..., x_m\}$, Consider $\phi(x) = [e^{\alpha x}, x e^{\alpha x}]$ and $\rho([u, v]) = (v - (v/u)e^{\alpha v/u})/(u - e^{\alpha v/u})$, then as $\alpha \to \infty$, we have $\rho(\phi(x_1) + \phi(x_2) + \phi(x_3) + ... + \phi(x_m))$ approaching the desired function.

# B  Proof of Lemma 3

Our goal is to design neural network layers that are equivariant to permutations of elements in the input $\mathbf{x}$. The function $\mathbf{f} : \mathfrak{X}^M \to \mathcal{Y}^M$ is **equivariant** to the permutation of its inputs iff

$$\mathbf{f}(\pi\mathbf{x}) = \pi\mathbf{f}(\mathbf{x}) \quad \forall \pi \in \mathcal{S}_M$$

where the symmetric group $\mathcal{S}_M$ is the set of all permutation of indices $1, \ldots, M$.

Consider the standard neural network layer

$$\mathbf{f}_\Theta(\mathbf{x}) \doteq \boldsymbol{\sigma}(\Theta\mathbf{x}) \quad \Theta \in \mathbb{R}^{M \times M} \tag{20}$$

where $\Theta$ is the weight vector and $\sigma : \mathbb{R} \to \mathbb{R}$ is a nonlinearity such as sigmoid function. The following lemma states the necessary and sufficient conditions for permutation-equivariance in this type of function.

**Lemma 3**  *The function $\mathbf{f}_\Theta : \mathbb{R}^M \to \mathbb{R}^M$ as defined in* (20) *is permutation equivariant if and only if all the off-diagonal elements of $\Theta$ are tied together and all the diagonal elements are equal as well. That is,*

$$\Theta = \lambda\mathbf{I} + \gamma\left(\mathbf{1}\mathbf{1}^\mathsf{T}\right) \qquad \lambda, \gamma \in \mathbb{R} \quad \mathbf{1} = [1, \ldots, 1]^\mathsf{T} \in \mathbb{R}^M$$

*where $\mathbf{I} \in \mathbb{R}^{M \times M}$ is the identity matrix.*

**Proof.**
From definition of permutation equivariance $\mathbf{f}_\Theta(\pi\mathbf{x}) = \pi\mathbf{f}_\Theta(\mathbf{x})$ and definition of $\mathbf{f}$ in (20), the condition becomes $\boldsymbol{\sigma}(\Theta\pi\mathbf{x}) = \pi\boldsymbol{\sigma}(\Theta\mathbf{x})$, which (assuming sigmoid is a bijection) is equivalent to $\Theta\pi = \pi\Theta$. Therefore we need to show that the necessary and sufficient conditions for the matrix $\Theta \in \mathbb{R}^{M \times M}$ to commute with all permutation matrices $\pi \in \mathcal{S}_M$ is given by this proposition. We prove this in both directions:

- To see why $\Theta = \lambda\mathbf{I} + \gamma\left(\mathbf{1}\mathbf{1}^\mathsf{T}\right)$ commutes with any permutation matrix, first note that commutativity is linear – that is

$$\Theta_1\pi = \pi\Theta_1 \wedge \Theta_2\pi = \pi\Theta_2 \quad \Rightarrow \quad (a\Theta_1 + b\Theta_2)\pi = \pi(a\Theta_1 + b\Theta_2).$$

  Since both Identity matrix $\mathbf{I}$, and constant matrix $\mathbf{1}\mathbf{1}^\mathsf{T}$, commute with any permutation matrix, so does their linear combination $\Theta = \lambda\mathbf{I} + \gamma\left(\mathbf{1}\mathbf{1}^\mathsf{T}\right)$.

- We need to show that in a matrix $\Theta$ that commutes with "all" permutation matrices
  - *All diagonal elements are identical*: Let $\pi_{k,l}$ for $1 \leq k, l \leq M, k \neq l$, be a transposition (*i.e.* a permutation that only swaps two elements). The inverse permutation matrix of $\pi_{k,l}$ is the permutation matrix of $\pi_{l,k} = \pi_{k,l}^\mathsf{T}$. We see that commutativity of $\Theta$ with the transposition $\pi_{k,l}$ implies that $\Theta_{k,k} = \Theta_{l,l}$:

$$\pi_{k,l}\Theta = \Theta\pi_{k,l} \Rightarrow \pi_{k,l}\Theta\pi_{l,k} = \Theta \Rightarrow (\pi_{k,l}\Theta\pi_{l,k})_{l,l} = \Theta_{l,l} \Rightarrow \Theta_{k,k} = \Theta_{l,l}$$

    Therefore, $\pi$ and $\Theta$ commute for any permutation $\pi$, they also commute for any transposition $\pi_{k,l}$ and therefore $\Theta_{i,i} = \lambda \, \forall i$.
  - *All off-diagonal elements are identical*: We show that since $\Theta$ commutes with any product of transpositions, any choice two off-diagonal elements should be identical. Let $(i, j)$ and $(i', j')$ be the index of two off-diagonal elements (*i.e.* $i \neq j$ and $i' \neq j'$). Moreover for now assume $i \neq i'$ and $j \neq j'$. Application of the transposition $\pi_{i,i'}\Theta$, swaps the rows $i, i'$ in $\Theta$. Similarly, $\Theta\pi_{j,j'}$ switches the $j^{th}$ column with $j'^{th}$ column. From commutativity property of $\Theta$ and $\pi \in \mathcal{S}_n$ we have

$$\pi_{j',j}\pi_{i,i'}\Theta = \Theta\pi_{j',j}\pi_{i,i'} \Rightarrow \pi_{j',j}\pi_{i,i'}\Theta(\pi_{j',j}\pi_{i,i'})^{-1} = \Theta \qquad\qquad \Rightarrow$$
$$\pi_{j',j}\pi_{i,i'}\Theta\pi_{i',i}\pi_{j,j'} = \Theta \Rightarrow (\pi_{j',j}\pi_{i,i'}\Theta\pi_{i',i}\pi_{j,j'})_{i,j} = \Theta_{i,j} \quad \Rightarrow \Theta_{i',j'} = \Theta_{i,j}$$

    where in the last step we used our assumptions that $i \neq i'$, $j \neq j'$, $i \neq j$ and $i' \neq j'$. In the cases where either $i = i'$ or $j = j'$, we can use the above to show that $\Theta_{i,j} = \Theta_{i'',j''}$ and $\Theta_{i',j'} = \Theta_{i'',j''}$, for some $i'' \neq i, i'$ and $j'' \neq j, j'$, and conclude $\Theta_{i,j} = \Theta_{i',j'}$.

∎

## C More Details on the architecture

### C.1 Invariant model

The structure of permutation invariant functions in Theorem 2 hints at a general strategy for inference over sets of objects, which we call deep sets. Replacing $\phi$ and $\rho$ by universal approximators leaves matters unchanged, since, in particular, $\phi$ and $\rho$ can be used to approximate arbitrary polynomials. Then, it remains to learn these approximators. This yields in the following model:

- Each instance $x_m \forall 1 \leq m \leq M$ is transformed (possibly by several layers) into some representation $\phi(x_m)$.
- The addition $\sum_m \phi(x_m)$ of these representations processed using the $\rho$ network very much in the same manner as in any deep network (*e.g.* fully connected layers, nonlinearities, *etc*).
- Optionally: If we have additional meta-information $z$, then the above mentioned networks could be conditioned to obtain the conditioning mapping $\phi(x_m|z)$.

Figure 5: Architecture of DeepSets: Invariant

In other words, the key to deep sets is to add up all representations and then apply nonlinear transformations.

The overall model structure is illustrated in Fig. 7.

This architecture has a number of desirable properties in terms of universality and correctness. We assume in the following that the networks we choose are, in principle, universal approximators. That is, we assume that they can represent any functional mapping. This is a well established property (see e.g. [50] for details in the case of radial basis function networks).

What remains is to state the derivatives with regard to this novel type of layer. Assume parametrizations $w_\rho$ and $w_\phi$ for $\rho$ and $\phi$ respectively. Then we have

$$\partial_{w_\phi} \rho \left( \sum_{x' \in X} \phi(x') \right) = \rho' \left( \sum_{x' \in X} \phi(x) \right) \sum_{x' \in X} \partial_{w_\phi} \phi(x')$$

This result reinforces the common knowledge of parameter tying in deep networks when ordering is irrelevant. Our result backs this practice with theory and strengthens it by proving that it is the only way to do it.

### C.2 Equivariant model

Consider the standard neural network layer

$$f_\Theta(\mathbf{x}) = \sigma(\Theta \mathbf{x}) \tag{21}$$

where $\Theta \in \mathbb{R}^{M \times M}$ is the weight vector and $\sigma : \mathbb{R}^M \to \mathbb{R}^M$ is a point-wise nonlinearity such as a sigmoid function. The following lemma states the *necessary and sufficient* conditions for permutation-equivariance in this type of function.

**Lemma 3** *The function $f_\Theta(\mathbf{x}) = \sigma(\Theta \mathbf{x})$ for $\Theta \in \mathbb{R}^{M \times M}$ is permutation equivariant, iff all the off-diagonal elements of $\Theta$ are tied together and all the diagonal elements are equal as well. That is,*

$$\Theta = \lambda \mathbf{I} + \gamma \left( \mathbf{1}\mathbf{1}^{\mathsf{T}} \right) \qquad \lambda, \gamma \in \mathbb{R} \quad \mathbf{1} = [1, \dots, 1]^{\mathsf{T}} \in \mathbb{R}^M$$

*where $\mathbf{I} \in \mathbb{R}^{M \times M}$ is the identity matrix.*

Figure 6: Illustration of permutation equivariant layer. Same color indicates weight sharing.

This function is simply a non-linearity applied to a weighted combination of i) its input $\mathbf{I}\mathbf{x}$ and; ii) the sum of input values $(\mathbf{1}\mathbf{1}^{\mathsf{T}})\mathbf{x}$. Since summation does not depend on the permutation, the layer is

permutation-equivariant. Therefore we can manipulate the operations and parameters in this layer, for example to get another **variation** $f(\mathbf{x}) = \sigma\left(\lambda\mathbf{I}\mathbf{x} + \gamma\,\text{maxpool}(\mathbf{x})\mathbf{1}\right)$, where the maxpooling operation over elements of the set (similarly to summation) is commutative. In practice using this variation performs better in some applications.

So far we assumed that each instance $x_m \in \mathbb{R}$ – *i.e.* a single input and also output channel. For **multiple input-output channels**, we may speed up the operation of the layer using matrix multiplication. For $D/D'$ input/output channels (*i.e.* $\mathbf{x} \in \mathbb{R}^{M \times D}$, $\mathbf{y} \in \mathbb{R}^{M \times D'}$, this layer becomes

$$f(\mathbf{x}) = \sigma\left(\mathbf{x}\Lambda - \mathbf{1}\mathbf{1}^{\mathsf{T}}\mathbf{x}\Gamma\right) \qquad (22)$$

Figure 7: Architecture of DeepSets: Equivariant

where $\Lambda, \Gamma \in \mathbb{R}^{D \times D'}$ are model parameters. As before, we can have a maxpool version as: $f(\mathbf{x}) = \sigma\left(\mathbf{x}\Lambda - \mathbf{1}\text{maxpool}(\mathbf{x})\Gamma\right)$ where $\text{maxpool}(\mathbf{x}) = (\max_m \mathbf{x}) \in \mathbb{R}^{1 \times D}$ is a row-vector of maximum value of $\mathbf{x}$ over the "set" dimension. We may further reduce the number of parameters in favor of better generalization by factoring $\Gamma$ and $\Lambda$ and keeping a single $\Lambda \in \mathbb{R}^{D,D'}$ and $\beta \in \mathbb{R}^{D'}$

$$f(\mathbf{x}) = \sigma\left(\beta + \left(\mathbf{x} - \mathbf{1}\text{maxpool}(\mathbf{x})\right)\Gamma\right) \qquad (23)$$

**Stacking:** Since composition of permutation equivariant functions is also permutation equivariant, we can build deep models by stacking layers of (23). Moreover, application of any commutative pooling operation (*e.g.* max-pooling) over the set instances produces a permutation *invariant* function.

Figure 8: Using multiple permutation equivariant layers. Since permutation equivariance compose we can stack multiple such layers

# D Bayes Set [36]

Bayesian sets consider the problem of estimating the likelihood of subsets $X$ of a ground set $\mathcal{X}$. In general this is achieved by an exchangeable model motivated by deFinetti's theorem concerning exchangeable distributions via

$$p(X|\alpha) = \int d\theta \left[ \prod_{m=1}^{M} p(x_m|\theta) \right] p(\theta|\alpha). \tag{24}$$

This allows one to perform set expansion, simply via the score

$$s(x|X) = \log \frac{p(X \cup \{x\}\,|\alpha)}{p(X|\alpha)p(\{x\}\,|\alpha)} \tag{25}$$

Note that $s(x|X)$ is the pointwise mutual information between $x$ and $X$. Moreover, due to exchangeability, it follows that regardless of the order of elements we have

$$S(X) := \sum_{m=1}^{M} s\left(x_m|\,\{x_{m-1},\ldots x_1\}\right) = \log p(X|\alpha) - \sum_{m=1}^{M} \log p(\{x_m\}\,|\alpha) \tag{26}$$

In other words, we have a set function $\log p(X|\alpha)$ with a modular term-dependent correction. When inferring sets it is our goal to find set completions $\{x_{m+1},\ldots x_M\}$ for an initial set of query terms $\{x_1,\ldots,x_m\}$ such that the aggregate set is well coherent. This is the key idea of the Bayesian Set algorithm.

## D.1 Exponential Family

In exponential families, the above approach assumes a particularly nice form whenever we have conjugate priors. Here we have

$$p(x|\theta) = \exp\left(\langle\phi(x),\theta\rangle - g(\theta)\right) \text{ and } p(\theta|\alpha, M_0) = \exp\left(\langle\theta,\alpha\rangle - M_0 g(\theta) - h(\alpha, M_0)\right). \tag{27}$$

The mapping $\phi : x \to \mathcal{F}$ is usually referred as sufficient statistic of $x$ which maps $x$ into a feature space $\mathcal{F}$. Moreover, $g(\theta)$ is the log-partition (or cumulant-generating) function. Finally, $p(\theta|\alpha, M_0)$ denotes the conjugate distribution which is in itself a member of the exponential family. It has the normalization $h(\alpha, M_0) = \int d\theta \exp\left(\langle\theta,\alpha\rangle - M_0 g(\theta)\right)$. The advantage of this is that $s(x|X)$ and $S(X)$ can be computed in closed form [36] via

$$s(X) = h\left(\alpha + \phi(X), M_0 + M\right) + (M-1)h(\alpha, M_0) - \sum_{m=1}^{M} h(\alpha + \phi(x_m), M+1) \tag{28}$$

$$\begin{aligned} s(x|X) = {}& h\left(\alpha + \phi(\{x\} \cup X), M_0 + M + 1\right) + h(\alpha, M_0) \\ & - h\left(\alpha + \phi(X), M_0 + M\right) - h(\alpha + \phi(x), M+1) \end{aligned} \tag{29}$$

For convenience we defined the sufficient statistic of a set to be the sum over its constituents, i.e. $\phi(X) = \sum_m \phi(x_m)$. It allows for very simple computation and maximization over additional elements to be added to $X$, since $\phi(X)$ can be precomputed.

## D.2 Beta-Binomial Model

The model is particularly simple when dealing with the Binomial distribution and its conjugate Beta prior, since the ratio of Gamma functions allows for simple expressions. In particular, we have

$$h(\beta) = \log\Gamma(\beta^+) + \log\Gamma(\beta^-) - \Gamma(\beta). \tag{30}$$

With some slight abuse of notation we let $\alpha = (\beta^+, \beta^-)$ and $M_0 = \beta^+ + \beta^-$. Setting $\phi(1) = (1,0)$ and $\phi(0) = (0,1)$ allows us to obtain $\phi(X) = (M^+, M^-)$, i.e. $\phi(X)$ contains the counts of occurrences of $x_m = 1$ and $x_m = 0$ respectively. This leads to the following score functions

$$\begin{aligned} s(X) = {}& \log\Gamma(\beta^+ + M^+) + \log\Gamma(\beta^- + M^-) - \log\Gamma(\beta + M) \\ & - \log\Gamma(\beta^+) - \log\Gamma(\beta^-) + \log\Gamma(\beta) - M^+ \log\frac{\beta^+}{\beta} - M^- \log\frac{\beta^-}{\beta} \end{aligned} \tag{31}$$

$$s(x|X) = \begin{cases} \log\frac{\beta^+ + M^+}{\beta + M} - \log\frac{\beta^+}{\beta} & \text{if } x = 1 \\ \log\frac{\beta^- + M^-}{\beta + M} - \log\frac{\beta^-}{\beta} & \text{otherwise} \end{cases} \tag{32}$$

This is the model used by [36] when estimating Bayesian Sets for objects. In particular, they assume that for any given object $x$ the vector $\phi(x) \in \{0; 1\}^d$ is a $d$-dimensional binary vector, where each coordinate is drawn independently from some Beta-Binomial model. The advantage of the approach is that it can be computed very efficiently while only maintaining minimal statistics of $X$.

In a nutshell, the *algorithmic* operations performed in the Beta-Binomial model are as follows:

$$s(x|X) = 1^\top \left[ \sigma \left( \sum_{m=1}^{M} \phi(x_m) + \phi(x) + \beta \right) - \sigma \left( \phi(x) + \beta \right) \right] \tag{33}$$

In other words, we sum over statistics of the candidates $x_m$, add a bias term $\beta$, perform a *coordinate-wise* nonlinear transform over the aggregate statistic (in our case a logarithm), and finally we aggregate over the so-obtained scores, weighing each contribution equally. $s(X)$ is expressed analogously.

### D.3 Gauss Inverse Wishart Model

Before abstracting away the probabilistic properties of the model, it is worth paying some attention to the case where we assume that $x_i \sim \mathcal{N}(\mu, \Sigma)$ and $(\mu, \Sigma) \sim \mathrm{NIW}(\mu_0, \lambda, \Psi, \nu)$, for a suitable set of conjugate parameters. While the details are (arguably) tedious, the overall structure of the model is instructive.

First note that the sufficient statistic of the data $x \in \mathbb{R}^d$ is now given by $\phi(x) = (x, xx^\top)$. Secondly, note that the conjugate log-partition function $h$ amounts to computing *determinants* of terms involving $\sum_m x_m x_m^\top$ and moreover, nonlinear combinations of the latter with $\sum_m x_m$.

The *algorithmic* operations performed in the Gauss Inverse Wishart model are as follows:

$$s(x|X) = \sigma \left( \sum_{m=1}^{M} \phi(x_m) + \phi(x) + \beta \right) - \sigma \left( \phi(x) + \beta \right) \tag{34}$$

Here $\sigma$ is a nontrivial convex function acting on a (matrix, vector) pair and $\phi(x)$ is no longer a trivial map but performs a nonlinear dimension altering transformation on $x$. We will use this general template to fashion the Deep Sets algorithm.

# E   Text Concept Set Retrieval

We consider the task of text concept set retrieval, where the objective is to retrieve words belonging to a 'concept' or 'cluster', given few words from that particular concept. For example, given the set of words {*tiger*, *lion*, *cheetah*}, we would need to retrieve other related words like *jaguar*, *puma*, *etc*, which belong to the same concept of big cats. The model implicitly needs to reason out the concept connecting the given set and then retrieve words based on their relevance to the inferred concept. Concept set retrieval is an important due to wide range of potential applications including personalized information retrieval, tagging large amounts of unlabeled or weakly labeled datasets, *etc*. This task of concept set retrieval can be seen as a set completion task conditioned on the latent semantic concept, and therefore our DeepSets form a desirable approach.

**Dataset**    To construct a large dataset containing sets of related words, we make use of Wikipedia text due to its huge vocabulary and concept coverage. First, we run topic modeling on publicly available wikipedia text with $K$ number of topics. Specifically, we use the famous latent Dirichlet allocation [38, 39], taken out-of-the-box[4]. Next, we choose top $N_T = 50$ words for each latent topic as a set giving a total of $K$ sets of size $N_T$. To compare across scales, we consider three values of $k = \{1k, 3k, 5k\}$ giving us three datasets LDA-1$k$, LDA-3$k$, and LDA-5$k$, with corresponding vocabulary sizes of $17k$, $38k$, and $61k$. Few of the topics from LDA-1k are visualized in Tab. 9.

**Methods**    Our DeepSets model uses a feedforward neural network (NN) to represent a query and each element of a set, *i.e.*, $\phi(x)$ for an element $x$ is encoded as a NN. Specifically, $\phi(x)$ represents each word via 50-dimensional embeddings that are we learn jointly, followed by two fully connected layers of size 150, with ReLU activations. We then construct a set representation or feature, by sum pooling all the individual representations of its elements, along with that of the query. Note that this sum pooling achieves permutation invariance, a crucial property of our DeepSets (Theorem 2). Next, use input this set feature into another NN to assign a single score to the set, shown as $\rho(.)$. We instantiate $\rho(.)$ as three fully connected layers of sizes $\{150, 75, 1\}$ with ReLU activations. In summary, our DeepSets consists of two neural networks – (a) to extract representations for each element, and (b) to score a set after pooling representations of its elements.

**Baselines**    We compare to several baselines: (a) **Random** picks a word from the vocabulary uniformly at random. (b) **Bayes Set** [36], and (c) **w2v-Near** that computes the nearest neighbors in the word2vec [40] space. Note that both Bayes Set and w2v NN are strong baselines. The former runs Bayesian inference using Beta-Binomial conjugate pair, while the latter uses the powerful 300 dimensional word2vec trained on the billion word GoogleNews corpus[5]. (d) **NN-max** uses a similar architecture as our DeepSets with an important difference. It uses max pooling to compute the set feature, as opposed to DeepSets which uses sum pooling. (e) **NN-max-con** uses max pooling on set elements but concatenates this pooled representation with that of query for a final set feature. (f) **NN-sum-con** is similar to NN-max-con but uses sum pooling followed by concatenation with query representation.

**Evaluation**    To quantitatively evaluate, we consider the standard retrieval metrics – recall@K, median rank and mean reciprocal rank. To elaborate, recall@K measures the number of true labels that were recovered in the top K retrieved words. We use three values of K = $\{10, 100, 1k\}$. The other two metrics, as the names suggest, are the median and mean of reciprocals of the true label ranks, respectively. Each dataset is split into TRAIN ($80\%$), VAL ($10\%$) and TEST ($10\%$). We learn models using TRAIN and evaluate on TEST, while VAL is used for hyperparameter selection and early stopping.

**Results and Observations**    Tab. 3 contains the results for the text concept set retrieval on LDA-1$k$, LDA-3$k$, and LDA-5$k$ datasets. We summarize our findings below: (a) Our /deepsets model outperforms all other approaches on LDA-3$k$ and LDA-5$k$ by any metric, highlighting the significance of permutation invariance property. For instance, /deepsets is better than the w2v-Near baseline by 1.5% in Recall@10 on LDA-5k. (b) On LDA-1$k$, neural network based models do not perform well when compared to w2v-Near. We hypothesize that this is due to small size of the dataset insufficient to train a high capacity neural network, while w2v-Near has been trained on a billion word corpus. Nevertheless, our approach comes the closest to w2v-Near amongst other approaches, and is only 0.5% lower by Recall@10.

| Topic 1 | Topic 2 | Topic 3 | Topic 4 | Topic 5 | Topic 6 |
|---------|---------|---------|---------|---------|---------|
| legend | president | plan | newspaper | round | point |
| airy | vice | proposed | daily | teams | angle |
| tale | served | plans | paper | final | axis |
| witch | office | proposal | news | played | plane |
| devil | elected | planning | press | redirect | direction |
| giant | secretary | approved | published | won | distance |
| story | presidency | planned | newspapers | competition | surface |
| folklore | presidential | development | editor | tournament | curve |

Figure 9: Examples from our LDA-1k datasets. Notice that each of these are latent topics of LDA and hence are semantically similar.

# F   Image Tagging

We next experiment with image tagging, where the task is to retrieve all relevant tags corresponding to an image. Images usually have only a subset of relevant tags, therefore predicting other tags can help enrich information that can further be leveraged in a downstream supervised task. In our setup, we learn to predict tags by conditioning /deepsets on the image. Specifically, we train by learning to predict a partial set of tags from the image and remaining tags. At test time, we the test image is used to predict relevant tags.

**Datasets**    We report results on the following three datasets:
(a) *ESPgame* [51]: Contains around $20k$ images spanning logos, drawings, and personal photos, collected interactively as part of a game. There are a total of $268$ unique tags, with each image having $4.6$ tags on average and a maximum of $15$ tags.
(b) *IAPRTC-12.5* [52]: Comprises of around $20k$ images including pictures of different sports and actions, photographs of people, animals, cities, landscapes, and many other aspects of contemporary life. A total of $291$ unique tags have been extracted from captions for the images. For the above two datasets, train/test splits are similar to those used in previous works [41, 44].
(c) *COCO-Tag:*  We also construct a dataset in-house, based on MSCOCO dataset[53]. COCO is a large image dataset containing around $80k$ train and $40k$ test images, along with five caption annotations. We extract tags by first running a standard spell checker[6] and lemmatizing these captions. Stopwords and numbers are removed from the set of extracted tags. Each image has $15.9$ tags on an average and a maximum of $46$ tags. We show examples of image tags from COCO-Tag in Fig. 10. The advantages of using COCO-Tag are three fold–richer concepts, larger vocabulary and more tags per image, making this an ideal dataset to learn image tagging using /deepsets.

**Image and Word Embeddings**    Our models use features extracted from Resnet, which is the state-of-the-art convolutional neural network (CNN) on ImageNet 1000 categories dataset using the publicly available 152-layer pretrained model[7]. To represent words, we jointly learn embeddings with the rest of /deepsets neural network for ESPgame and IAPRTC-12.5 datasets. But for COCO-Tag, we bootstrap from $300$ dimensional word2vec embeddings[8] as the vocabulary for COCO-Tag is significantly larger than both ESPgame and IAPRTC-12.5 ($13k$ vs $0.3k$).

**Methods**    The setup for DeepSets to tag images is similar to that described in Appendix E. The only difference being the conditioning on the image features, which is concatenated with the set feature obtained from pooling individual element representations. In particular, $\phi(x)$ represents each word via 300-dimensional word2vec embeddings, followed by two fully connected layers of size 300, with ReLU activations, to construct the set representation or features. As mentioned earlier, we concatenate the image features and pass this set features into another NN to assign a single score to the set, shown as $\rho(.)$. We instantiate $\rho(.)$ as three fully connected layers of sizes $\{300, 150, 1\}$ with ReLU activations. The resulting feature forms the new input to a neural network used to score the set, in this case, score the relevance of a tag to the image.

**Baselines**    We perform comparisons against several baselines, previously reported from [41]. Specifically, we have Least Sq., a ridge regression model, MBRM [42], JEC [43] and FastTag [41]. Note that these methods do not use deep features for images, which could lead to an unfair comparison. As there is no publicly available code for MBRM and JEC, we cannot get performances of these models with Resnet extracted features. However, we report results with deep features for FastTag and Least Sq., using code made available by the authors [9].

**Evaluation** For ESPgame and IAPRTC-12.5, we follow the evaluation metrics as in [44] – precision (P), recall (R), F1 score (F1) and number of tags with non-zero recall (N+). Note that these metrics are evaluate for each tag and the mean is reported. We refer to [44] for further details. For COCO-Tag, however, we use recall@K for three values of K = $\{10, 100, 1000\}$, along with median rank and mean reciprocal rank (see evaluation in Appendix E for metric details).

**Results and Observations** Tab. 4 contains the results of image tagging on ESPgame and IAPRTC-12.5, and Tab. 5 on COCO-Tag. Here are the key observations from Tab. 4: (a) The performance of /deepsets is comparable to the best of other approaches on all metrics but precision. (b) Our recall beats the best approach by 2% in ESPgame. On further investigation, we found that /deepsets retrieves more relevant tags, which are not present in list of ground truth tags due to a limited 5 tag annotation. Thus, this takes a toll on precision while gaining on recall, yet yielding improvement in F1. On the larger and richer COCO-Tag, we see that /deepsets approach outperforms other methods comprehensively, as expected. We show qualitative examples in Fig. 10.

| GT | Pred | GT | Pred | GT | Pred |
|---|---|---|---|---|---|
| building | building | standing | person | traffic | clock |
| sign | street | surround | group | city | tower |
| brick | city | woman | man | building | sky |
| picture | brick | crowd | table | tall | building |
| empty | sidewalk | wine | sit | large | tall |
| white | side | person | room | tower | large |
| black | pole | group | woman | European | cloudy |
| street | white | table | couple | front | front |
| image | stone | bottle | gather | clock | city |

| GT | Pred | GT | Pred | GT | Pred |
|---|---|---|---|---|---|
| photograph | ski | laptop | refrigerator | beach | jet |
| snowboarder | snow | person | fridge | shoreline | airplane |
| snow | slope | screen | room | stand | propeller |
| glide | person | room | magnet | walk | ocean |
| hill | snowy | desk | cabinet | sand | plane |
| show | hill | living | kitchen | lifeguard | water |
| person | man | counter | shelf | white | body |
| slope | skiing | computer | wall | person | person |
| young | skier | monitor | counter | surfboard | sky |

Figure 10: Qualitative examples of image tagging using /deepsets. *Top row*: Positive examples where most of the retrieved tags are present in the ground truth (brown) or are relevant but not present in the ground truth (green). *Bottom row*: Few failure cases with irrelevant/wrong tags (red). From left to right, (i) Confusion between snowboarding and skiing, (ii) Confusion between back of laptop and refrigerator due to which other tags are kitchen-related, (iii) Hallucination of airplane due to similar shape of surfboard.

# G   Improved Red-shift Estimation Using Clustering Information

An important regression problem in cosmology is to estimate the red-shift of galaxies, corresponding to their age as well as their distance from us [33]. Two common types of observation for distant galaxies include a) photometric and b) spectroscopic observations, where the latter can produce more accurate red-shift estimates.

One way to estimate the red-shift from photometric observations is using a regression model [34]. We use a multi-layer Perceptron for this purpose and use the more accurate spectroscopic red-shift estimates as the ground-truth. As another baseline, we have a photometric redshift estimate that is provided by the catalogue and uses various observations (including clustering information) to estimate individual galaxy-red-shift. Our objective is to use clustering information of the galaxies to improve our red-shift prediction using the multi-layer Preceptron.

Note that the prediction for each galaxy does not change by permuting the members of the galaxy cluster. Therefore, we can treat each galaxy cluster as a "set" and use permutation-equivariant layer to estimate the individual galaxy red-shifts.

For each galaxy, we have 17 photometric features [10] from the redMaPPer galaxy cluster catalog [35], which contains photometric readings for 26,111 red galaxy clusters. In this task in contrast to the previous ones, sets have different cardinalities; each galaxy-cluster in this catalog has between $\sim 20-300$ galaxies – *i.e.* $\mathbf{x} \in \mathbb{R}^{N(c) \times 17}$, where $N(c)$ is the cluster-size. See Fig. 11(a) for distribution of cluster sizes. The catalog also provides accurate spectroscopic red-shift estimates for a *subset* of these galaxies as well as photometric estimates that uses clustering information. Fig. 11(b) reports the distribution of available spectroscopic red-shift estimates per cluster.

We randomly split the data into 90% training and 10% test clusters, and use the following simple architecture for semi-supervised learning. We use four permutation-equivariant layers with 128, 128, 128 and 1 output channels respectively, where the output of the last layer is used as red-shift estimate. The squared loss of the prediction for available spectroscopic red-shifts is minimized.[11] Fig. 11(c) shows the agreement of our estimates with spectroscopic readings on the galaxies in the test-set with spectroscopic readings. The figure also compares the photometric estimates provided by the catalogue [35], to the ground-truth. As it is customary in cosmology literature, we report the average **scatter** $\frac{|z_{\text{spec}} - z|}{1 + z_{\text{spec}}}$, where $z_{\text{spec}}$ is the accurate spectroscopic measurement and $z$ is a photometric estimate. The average scatter using **our model** is .023 compared to the scatter of .025 in the **original photometric estimates** for the redMaPPer catalog. Both of these values are averaged over all the galaxies with spectroscopic measurements in the test-set.

We repeat this experiment, replacing the permutation-equivariant layers with fully connected layers (with the same number of parameters) and only use the individual galaxies with available spectroscopic estimate for training. The resulting average scatter for **multi-layer Perceptron** is .026, demonstrating that using clustering information indeed improves photometric red-shift estimates.

Figure 11: Application of permutation-equivariant layer to semi-supervised red-shift prediction using clustering information: (a) distribution of cluster (set) size; (b) distribution of reliable red-shift estimates per cluster; (c) prediction of red-shift on test-set (versus ground-truth) using clustering information as well as RedMaPPer photometric estimates (also using clustering information).

Figure 12: Examples for 8 out of 40 object classes (column) in the ModelNet40. Each point-cloud is produces by sampling 1000 particles from the mesh representation of the original MeodelNet40 instances. Two point-clouds in the same column are from the same class. The projection of particles into xy, zy and xz planes are added for better visualization.

## H   Point Cloud Classification

Tab. 6 presents a more detailed result on classification performance, using different techniques. Fig. 12 shows examples of the dataset used for training. Fig. 13 shows the features learned by the first and second layer of our deep model. Here, we review the details of architectures used in the experiments.

**DeepSets** We use a network comprising of 3 permutation-equivariant layers with 256 channels followed by max-pooling over the set structure. The resulting vector representation of the set is then fed to a fully connected layer with 256 units followed by a 40-way softmax unit. We use Tanh activation at all layers and dropout on the layers after set-max-pooling (*i.e.* two dropout operations) with 50% dropout rate. Applying dropout to permutation-equivariant layers for point-cloud data deteriorated the performance. We observed that using different types of permutation-equivariant layers (see Appendix C) and as few as 64 channels for set layers changes the result by less than 5% in classification accuracy.

For the setting with 5000 particles, we increase the number of units to 512 in all layers and randomly rotate the input around the $z$-axis. We also randomly scale the point-cloud by $s \sim \mathcal{U}(.8, 1./.8)$. For this setting only, we use Adamax [54] instead of Adam and reduce learning rate from .001 to .0005.

**Graph convolution.** For each point-cloud instance with 1000 particles, we build a sparse K-nearest neighbor graph and use the three point coordinates as input features. We normalized all graphs at the preprocessing step. For direct comparison with set layer, we use the exact architecture of 3 graph-convolution layer followed by set-pooling (global graph pooling) and dense layer with 256 units. We use exponential linear activation function instead of Tanh as it performs better for graphs. Due to over-fitting, we use a heavy dropout of 50% after graph-convolution and dense layers. Similar to dropout for sets, all the randomly selected features are simultaneously dropped across the graph nodes. the We use a mini-batch size of 64 and Adam for optimization where the learning rate is .001 (the same as that of permutation-equivariant counter-part).

Despite our efficient sparse implementation using Tensorflow, graph-convolution is significantly slower than the set layer. This prevented a thorough search for hyper-parameters and it is quite possible that better hyper-parameter tuning would improve the results that we report here.

Table 6: Classification accuracy and the (size of) representation used by different methods on the ModelNet40 dataset.

| model | instance size | representation | accuracy |
|---|---|---|---|
| **DeepSets** + transformation (ours) | $5000 \times 3$ | point-cloud | $90 \pm .3\%$ |
| **DeepSets** (ours) | $1000 \times 3$ | point-cloud | $87 \pm 1\%$ |
| **Deep-Sets w. pooling only** (ours) | $1000 \times 3$ | point-cloud | $83 \pm 1\%$ |
| **DeepSets** (ours) | $100 \times 3$ | point-cloud | $82 \pm 2\%$ |
| KNN graph-convolution (ours) | $1000 \times (3 + 8)$ | directed 8-regular graph | $58 \pm 2\%$ |
| 3DShapeNets [25] | $30^3$ | voxels (using convolutional deep belief net) | $77\%$ |
| DeepPano [20] | $64 \times 160$ | panoramic image (2D CNN + angle-pooling) | $77.64\%$ |
| VoxNet [26] | $32^3$ | voxels (voxels from point-cloud + 3D CNN) | $83.10\%$ |
| MVCNN [21] | $164 \times 164 \times 12$ | multi-vew images (2D CNN + view-pooling) | $90.1\%$ |
| VRN Ensemble [27] | $32^3$ | voxels (3D CNN, variational autoencoder) | $95.54\%$ |
| 3D GAN [28] | $64^3$ | voxels (3D CNN, generative adversarial training) | $83.3\%$ |

Tab. 6 compares our method against the competition.[12] Note that we achieve our best accuracy using $5000 \times 3$ dimensional representation of each object, which is much smaller than most other methods. All other techniques use either voxelization or multiple view of the 3D object for classification. Interestingly, variations of view/angle-pooling, as in [20, 21], can be interpreted as set-pooling where the class-label is invariant to permutation of different views. The results also shows that using fully-connected layers with set-pooling alone (without max-normalization over the set) works relatively well.

We see that reducing the number of particles to only 100, still produces comparatively good results. Using graph-convolution is computationally more challenging and produces inferior results in this setting. The results using 5000 particles is also invariant to small changes in scale and rotation around the $z$-axis.

Units of the **first** permutation-invariant layer

Units of the **second** permutation-invariant layer

Figure 13: Each box is the particle-cloud maximizing the activation of a unit at the firs (**top**) and second (**bottom**) permutation-equivariant layers of our model. Two images of the same column are two different views of the same point-cloud.

**Features.** To visualize the features learned by the set layers, we used Adamax [54] to locate 1000 particle coordinates maximizing the activation of each unit.[13] Activating the tanh units beyond the second layer proved to be difficult. 13 shows the particle-cloud-features learned at the first and second layers of our deep network. We observed that the first layer learns simple localized (often cubic) point-clouds at different $(x, y, z)$ locations, while the second layer learns more complex surfaces with different scales and orientations.

# I    Set Anomaly Detection

Our model has 9 convolution layers with $3 \times 3$ receptive fields. The model has convolution layers with $32, 32, 64$ feature-maps followed by max-pooling followed by 2D convolution layers with $64, 64, 128$ feature-maps followed by another max-pooling layer. The final set of convolution layers have $128, 128, 256$ feature-maps, followed by a max-pooling layer with pool-size of 5 that reduces the output dimension to batch-size.$M \times 256$, where the set-size $M = 16$. This is then forwarded to three permutation-equivariant layers with $256, 128$ and 1 output channels. The output of final layer is fed to the Softmax, to identify the outlier. We use exponential linear units [55], drop out with 20% dropout rate at convolutional layers and 50% dropout rate at the first two set layers. When applied to set layers, the selected feature (channel) is simultaneously dropped in all the set members of that particular set. We use Adam [54] for optimization and use batch-normalization only in the convolutional layers. We use mini-batches of 8 sets, for a total of 128 images per batch.

Figure 14: *Each row shows a set, constructed from CelebA dataset, such that all set members except for an outlier, share at least two attributes (on the right). The **outlier is identified with a red frame**. The model is trained by observing examples of sets and their anomalous members, **without access to the attributes**. The probability assigned to each member by the outlier detection network is visualized using a **red bar** at the bottom of each image.*

Figure 15: *Each row of the images shows a set, constructed from CelebA dataset images, such that all set members except for an outlier, share at least two attributes. The **outlier is identified with a red frame**. The model is trained by observing examples of sets and their anomalous members and **without access to the attributes**. The probability assigned to each member by the outlier detection network is visualized using a **red bar** at the bottom of each image. The probabilities in each row sum to one.*

## Footnotes

[4] `github.com/dmlc/experimental-lda`

[5] `code.google.com/archive/p/word2vec/`

[6] http://hunspell.github.io/

[7] github.com/facebook/fb.resnet.torch

[8] https://code.google.com/p/word2vec/

[9] http://www.cse.wustl.edu/~mchen/

[10]We have a single measurement for each u,g,r, i and z band as well as measurement error bars, location of the galaxy in the sky, as well as the probability of each galaxy being the cluster center. We do not include the information regarding the richness estimates of the clusters from the catalog, for any of the methods, so that baseline multi-layer Preceptron is blind to the clusters.

[11]We use mini-batches of size 128, Adam [54], with learning rate of .001, $\beta_1 = .9$ and $\beta_2 = .999$. All layers except for the last layer use Tanh units and simultaneous dropout with 50% dropout rate.

[12]The error-bar on our results is due to variations depending on the choice of particles during test time and it is estimated over three trials.

[13]We started from uniformly distributed set of particles and used a learning rate of .01 for Adamax, with first and second order moment of .1 and .9 respectively. We optimized the input in $10^5$ iterations. The results of Fig. 13 are limited to instances where tanh units were successfully activated. Since the input at the first layer of our deep network is normalized to have a zero mean and unit standard deviation, we do not need to constrain the input while maximizing unit's activation.