[Reviews · NeurIPS 2017]

Reviewer 1



The paper presents a way to design neural networks that are invariant / equivariant to permutations, making them suitable for set-valued data. A proof is given that essentially any invariant function can be written in the proposed form rho(sum_x phi(x)), where phi is a learned feature transform and rho processes the summed representations. Furthermore, such a function is always invariant. The paper contains extensive experiments, that show the proposed architecture is applicable to a wide range of problems and yields competitive results across the board. The experiment on digit addition shows very good performance, but it should be noted that addition is directly baked into the model, so this is perhaps not very surprising. The paper is fairly well written and very well structured. Section 2.3 discusses well known related results. It took me quite a while to understand what the relation to the present work is. I think this section could be made clearer. Lemma 3 is correct, but implicitly assumes that the representation of the symmetric group acting on the input should be the same representation as the one acting on the output. A simple example of an alternative equivariant map could be one where the output is a scalar that changes sign when an odd permutation is applied and otherwise stays the same. Such a map could do pairwise comparisons at each layer, which I suspect would improve results for e.g. point cloud prediction. Lemma 3 was derived before in “Equivariance through parameter sharing” and other papers by Ravanbakhsh. The paper by Ravanbakhsh on “estimating cosmological …” appears twice in the reference list. In figure 1, why does the curve for SDM stop before the DeepSets curve? In summary, I think this is a solid paper with a strong theoretical foundation and extensive experimental validation. Minor comments: 3: “and are” -> “that are” 30: “in field” -> “in the field” 47: “not to be indifferent” -> “to be indifferent” 50-57: several typos / grammar mistakes 84: it is not immediately clear what M_0 is; it would be good to have a line about it. 156: matrices

Reviewer 2



The paper characterises functions that are permutation invariant and equivariant (the latter under certain restrictions); it then derives form these characterisation principle for the design of deep neural networks that operate on set-like objects. A number of diverse experiments validate the approach. The problem of learning to classify or regress from sets is an important one. The diversity of applications considered in the experiments is notable, although some of them are "mere" toy problems. Nevertheless, I was still slightly turned off by the nature of some of the results. Theorem 2 indicates that any function of the power set of a countable set has an additive decomposition (by the way, it is never said that the functions $Phi$ in the decomposition can be a scalar one, but that is the case from the appendix). On a glance, this is an exciting result. However, it is obtained by constructing a surjective mapping from the power set of N to the real numbers. This construction does not show that such an additive decomposition is a reasonable one in a learning setting. For example, representing the function that computes the maximum element in a set (which is obviously permutation invariant) may be very inconvenient to code in this manner. Furthermore, the proof itself may be a bit problematic as the proposed mapping does not seem to be surjective after all; for example, 2^-1 = sum_{k=2}^{+inf} 2^-k, so the sets {1,0,0,0,...} and {0,1,1,1,1,...} would have the same code. Lemma 3 is also very restrictive. Here the authors show that a fully connected layer which multiplies all input components by the same scalar and sums to that a scaled average of the components is the only one that can be permutation equivariant. This is fine, but at the same time does not allow the neural network to do much. Most importantly, these two theoretical results do no appear to suggest genuinely new approaches to permutation-invariant data processing in neural networks. The idea of using commutative pooling operators (sum or pooling), in particular, is completely standard. The sum of digits experiment is a bit suspicious. In the text case, in particular, it would be enough to map each digit to its corresponding numerical value (a triviality) to allow the sum operator solve the problem exactly -- thus is not exactly surprising that this method works better than LSTM and GRU in this case. The image case is only slightly harder. The rebuttal is informative and answers the most important questions raised in the review well. Therefore I am slightly upgrading my already positive score. By the way, I understand the necessary condition nature of Lemma 3. My point is that this results also shows that there isn't all that much that can be done in terms of data processing if one wants to be permutation invariant.

Reviewer 3



Summary of Review: Although the techniques are not novel, the proofs are, and the paper seems well done overall. I’m not familiar with the experiments / domains of the tasks, and there’s not enough description for me to confidently judge the experiments, but they appear fairly strong. Also, the authors missed one key piece of related work (“Learning Multiagent Communication with Backpropagation” - Sukhbaatar et al. 2016). Overview of paper: This paper examines permutation-invariant/equivariant deep learning, characterizing the class of functions (on countable input spaces) and DNN layers which exhibit these properties. There is also an extensive experimental evaluation. The paper is lacking in clarity of presentation and this may hide some lack of conceptual clarity as well; for instance, in some of the applications given (e.g. anomaly detection), the input is properly thought of as a multiset, not a set. The proof of Theorem 2 would not apply to multisets. Quality: Overall, the paper appears solid in motivation, theory, and experiments. The proofs (which I view as the main contribution) are not particularly difficult, impressive, or involved, but they do provide valuable results. The other contributions are a large number of experiments, which I have difficulty judging. The take-aways of the experiments are fairly clear: 1) expanding deep learning to solve set-based tasks can (yet again) yield improvements over approaches that don’t use deep learning 2) in many cases, a naive application of deep learning is bound to fail (for instance, section 4.3, where the position of the anomaly is, presumably, random, and thus impossible for a permutation *invariant* architecture, which I presume the baseline to be), and permutation invariance / equivariance *must* be considered and properly incorporated into the model design. Clarity: While the overall message is clear, the details are not, and I’ll offer a number of corrections / suggestions, which I expect the authors to address or consider. First of all, notation should be defined (for instance, this is missing in de Finetti’s theorem). Section 2.3 should, but does not explicitly show how these results relate to the rest of the paper. Line 136-137: you should mention some specific previous applications, and be clear, later on in the experiments section, about the novelty of each experiment. Line 144: which results are competitive with state-of-the-art (SOTA)? Please specify, and reference previous SOTA. As mentioned before, the tasks are not always explained clearly enough for someone unfamiliar with them (not even in the appendix!). In particular, I had difficulty understanding the tasks in section 4.1.3 and 4.2. In 4.2 I ask: How is “set expansion” performed? What does “coherent” mean? Is the data labelled (e.g. with ground-truth set assignments?) What is the performance metric? What is the “Beta-Binomial model”? detailed comments/questions: There are many missing articles, e.g. line 54 “in transductive” —> “in the transductive” The 2nd sentence of the abstract is not grammatical. line 46: “not” should be removed line 48: sets do not have an ordering; they are “invariant by definition”. Please rephrase this part to clarify (you could talk about “ordered sets”, for instance). Line 151: undefined notation (R(alpha), M). Lines 161-164 repeat the caption of Figure 1; remove/replace duplicate text. the enumeration in lines 217-273 is poorly formatted Line 275 d) : is this the *only* difference with your method? Max-pooling is permutation invariant, so this would still be a kind of Deep Set method, right?? line 513: the mapping must be a bijection for the conclusions to follow! As I mentioned, set and multiset should be distinguished; a multiset is permutation invariant but may have repeated elements. Multisets may be more appropriate for some machine learning tasks. BTW, I suspect the result of Theorem 2 would not hold for uncountable sets. Bayesian Sets is cited twice (35 and 49). Originality: I’m a bit out of my depth in terms of background on the tasks in this paper, and I’m not 100% sure the proofs are novel; I would not be particularly surprised if these results are known in, e.g. some pure mathematics communities. My best guess is that the proofs are novel, and I’m fairly confident they’re novel to the machine learning or at least deep learning community. Significance: I think the proofs are significant and likely to clarify the possible approaches to these kinds of set-based tasks for many readers. As a result (and also due to the substantial experimental work), the paper may also play a role in popularizing the application of deep nets to set-based tasks, which would be quite significant.